# Memory-Efficient Fine-Tuning via Structured Neural Network Pruning

## Abstract

Fine-tuning is an important step in adapting foundation models such as large language models to downstream tasks. To make this step more accessible to users with limited computational budgets, it is crucial to develop fine-tuning methods that are memory and computationally efficient. Sparse Fine-tuning (SFT) and Low-rank adaptation (LoRA) are two frameworks that have emerged for addressing this problem, and have been adopted widely in practice. In this work, we develop a new SFT framework, based on ideas from neural network pruning. At a high level, we first identify "important" neurons/nodes using feature importance metrics from network pruning (specifically, we use the structural pruning method), and then perform fine-tuning by restricting to weights involving these neurons. Using experiments on both vision and language tasks, we demonstrate that our method significantly improves the memory efficiency of SFT without increasing training time complexity and implementation complexity, while achieving accuracy comparable to state-of-the-art methods such as LoRA and its variants.

## 1 Introduction

The paradigm of *pre-training followed by fine-tuning* has seen tremendous success in the last few years. Very large models (often referred to as foundation models) are first trained, typically using very large amounts of data and computational resources, using self-supervised learning approaches (Dosovitskiy, 2020; Achiam et al., 2023; Dubey et al., 2024; Zhou et al., 2024). When building a model for a new task (which could be a supervised learning task), the idea is to start with the foundation model and then tune its parameters, possibly after adding additional classification layers, by training using task-specific data. The pre-train then fine-tune paradigm has been shown to have significant advantages over training a new model from scratch for the new task. Often, high accuracy can be obtained using much smaller datasets for the new task.

Despite the success, fine-tuning a model with billions of parameters requires access to heavy computational resources, even when the task datasets are fairly small. Fortunately, it has been observed (e.g., see (Panigrahi et al., 2023) and references therein) that fine-tuning can often be done by tuning only a small fraction of the model parameters. Parameter-efficient fine-tuning (PEFT) methods have thus been proposed to carry out this idea and deal with the challenge of making fine-tuning more accessible (Lialin et al., 2023). Commonly used PEFT methods include Low-Rank Adaptation (LoRA, Hu et al. 2022) sparse fine-tuning (SFT, Sung et al. 2021; Guo et al. 2021; Ansell et al. 2022; Nikdan et al. 2024). LoRA, the most widely used PEFT, achieves memory efficiency by simply making low-rank updates to the weight matrices in the different layers. In contrast, SFT learns a sparse matrix for updates, typically an unstructured one. Due to this lack of structure, SFT methods typically have a higher memory usage during the fine-tuning process than LoRA. As the scale of LLMs increases, continuing to advance the field of PEFT is essential, and there has thus been a large body of work towards making progress (see also methods such as Malladi et al. (2023)).

The methods above for fine-tuning resemble the literature on neural network compression or "network pruning" (Han et al., 2015; Han, 2017). This line of work, starting with the seminal paper of LeCun et al. (1989), aims to develop smaller models that have the same functional behavior as a much larger neural network. The primary applications are in deploying NN models on edge devices that are power- and resource- constrained. Ideas such as low rank factorization and sparsity, combined with quantization (representing the weights using 8-bit or 4-bit data types; e.g., see (Gholami et al., 2022))

have played a key role in NN compression. Another prominent class of methods are unstructured and structured pruning (Cheng et al., 2024). The former zeros out less important parameters (resulting in a sparse weight matrix), while structured pruning removes the least important neurons or channels (resulting in a smaller dimensional weight matrix). Both reduce the model's space and computational complexity without significantly degrading accuracy. Despite similarity in methods, to our knowledge, pruning techniques have not been directly useful in model fine-tuning.

Although numerous works have explored unstructured pruning and SFT, a key challenge persists: unstructured sparse matrices require additional implementations for the training process to achieve memory efficiency. This often involves optimizing tensor computations by selectively processing only non-zero elements, e.g. torch.sparse[1], compressed sparse column/row (CSC/CSR, Mofrad et al., 2019), semi-structured formats (Holmes et al., 2021), etc. The tradeoff in these approaches lies in the fact that they all increase time complexity to achieve reduced memory complexity. Therefore, some approaches also leverage C++ for acceleration, as seen in works like (Nikdan et al., 2024; 2023)[2]. The necessity for such additional implementations complicates the practical application of these methods and increases the difficulty of further advancing this field.

In this work, we study the question: *Can sparse fine-tuning be improved by incorporating techniques from neural network compression and matrix decomposition to create a memory- and parameter-efficient framework, while avoiding additional implementations of sparse operations and without increasing the training time complexity?* We answer this question in the affirmative, by proposing a new SFT framework for fine-tuning LLMs and Vision Transformers that achieves memory- and parameter-efficiency while maintaining or even improving performance on downstream tasks. Our approach utilizes structured NN pruning to identify a subset of fine-tuning parameters and employs a matrix decomposition-based computation for efficient fine-tuning. This design enables the integration of ideas from model compression, SFT, and matrix decomposition methods.

The rest of the paper is organized as follows. We outline our contributions in Section 1.1. We then discuss existing PEFT methods in Section 2 and describe our approach in detail in Section 3. Section 4 describes the settings of our experiments. We then present and discuss our results in Section 5. We also analyze the memory efficiency of our method. Section 6 concludes with some directions for future work along our lines.

## 1.1 Our Contributions

At a high level, our contributions are as follows:

- We enhance SFT by combining network pruning and matrix decomposition, achieving significant memory efficiency. Our method uses standard tensor operations, eliminating the need for custom implementations for sparse tensors.

- By replacing unstructured sparse matrices with structured ones, we achieve memory efficiency lower than the popular LoRA with comparable trainable parameters. Our modular approach enables the integration of pruning techniques for neuron importance and works with all layer types, including LayerNorm and BatchNorm, which LoRA cannot directly handle.

- We validate our method across diverse fine-tuning tasks (language and vision) and provide practical guidance on hyperparameter and training configuration selection to maximize efficiency and accuracy.

## 2 Background and Related Work

**Parameter-Efficient and Memory-Efficient Fine-Tuning**: In various language and vision tasks, the "pre-train then fine-tune" paradigm has been shown highly effective. PEFT methods fine-tune a small

---

[1]The beta version of torch.sparse please see https://pytorch.org/docs/stable/sparse.html

[2]The implementation of their forward pass, backward pass, and back-propagation can be found in https://github.com/IST-DASLab/spops, where the tensor operations are mostly implemented by C++.

subset of the parameters of a large pre-trained model in order to accelerate the training process. We begin by introducing SFT and LoRA, two popular approaches for PEFT.

**Sparse Fine-Tuning**: SFT formulates the fine-tuning process as learning another weight matrix $\mathbf{W}_s$:

$$\hat{\mathbf{W}} = \mathbf{W} + \mathbf{W}_s, \tag{1}$$

$$\mathbf{h} = f(\hat{\mathbf{W}}, \mathbf{x}) = f(\mathbf{W} + \mathbf{W}_s, \mathbf{x}), \tag{2}$$

where $\mathbf{h} \in \mathbb{R}^{d_{out}}$ and $\mathbf{x} \in \mathbb{R}^{d_{in}}$ are the input and output of the hidden layer, respectively, $f(\cdot)$ is the forward function, $\mathbf{W} \in \mathbb{R}^{d_{out} \times d_{in}}$ represents the frozen pre-trained parameters, and $\hat{\mathbf{W}} \in \mathbb{R}^{d_{out} \times d_{in}}$ denotes the final parameters used during inference for the fine-tuning task. The matrix $\mathbf{W}_s \in \mathbb{R}^{d_{out} \times d_{in}}$ is sparse and is initialized at $\mathbf{0}$. Such a decomposition is done for every layer in the neural network. SFT methods try to limit the number of parameters to fine-tune. For selecting non-zero indices, Guo et al. (2021) propose learning a mask for $\mathbf{W}_s$ (using a standard Backprop algorithm), while Sung et al. (2021) uses Fisher information to identify important indices in $\mathbf{W}$. Ansell et al. (2022) fine-tune the whole model for one epoch, then use $\mathbf{W}_s$ itself as an importance score to decide which parameters to fine-tune subsequently. However, the key challenge of all SFT methods is that they do not sufficiently reduce memory usage, as $\mathbf{W}_s$ keeps the dimensionality of $\mathbf{W}$, and thus standard libraries do not yield memory improvements.

**Techniques for Memory Efficient Training**: To address memory redundancy when computing sparse tensors, various data formats, such as compressed sparse column/row (CSC/CSR, Mofrad et al., 2019; Lu et al., 2024) and semi-structured formats (Holmes et al., 2021) are proposed. These formats enable efficient element-wise operations like Sparse Matrix Multiplication (SpMM), which is crucial for performing dot products and matrix multiplications efficiently. Upon these techniques, sparse backpropagation is built to efficiently train models (Zhang et al., 2020; Gale et al., 2020; Peste et al., 2021; Schwarz et al., 2021; Hoefler et al., 2021; Jiang et al., 2022; Nikdan et al., 2023; Xu et al., 2024). Beyond sparse tensor techniques, NVIDIA also offers memory optimization techniques for efficient training, see their memory optimizations page[3].

However, these techniques come with trade-offs, particularly in terms of time complexity and implementation complexity. Achieving memory efficiency often requires a significant increase in time complexity. To mitigate this, some approaches employ optimizations implemented in C++ or lower-level languages, such as those used in (Gale et al., 2020; Nikdan et al., 2023; 2024), to accelerate the training process.

**Low-Rank Adaptation (LoRA)**: Instead of requiring $\mathbf{W}_s$ to be sparse, low-rank adaptation aims to find update matrices that are of small rank:

$$\hat{\mathbf{W}} = \mathbf{W} + \frac{\alpha}{r}\mathbf{BA} \tag{3}$$

$$\mathbf{h} = f(\hat{\mathbf{W}}, \mathbf{x}) = f(\mathbf{W} + \frac{\alpha}{r}\mathbf{BA}, \mathbf{x}) = f(\mathbf{W}, \mathbf{x}) + f(\frac{\alpha}{r}\mathbf{BA}, \mathbf{x}), \tag{4}$$

where $\alpha$ is the LoRA scaling hyper-parameter, $\mathbf{B} \in \mathbb{R}^{d_{out} \times r}$, $\mathbf{A} \in \mathbb{R}^{r \times d_{in}}$ are the low-rank matrices with $r \ll d_{in}, d_{out}$. During inference, the $\mathbf{BA}$ term can be merged into $\mathbf{W}$ to maintain the inference latency of the original model. During training, owing to the fact that $f$ is additive for both the self-attention blocks and the subsequent multilayer perceptron (MLP) layers of transformers (Vaswani, 2017), backpropagation can be performed efficiently for the $\mathbf{B}, \mathbf{A}$ parameters. Due to LoRA's simplicity and effectiveness, numerous variants have been proposed to enhance the performance.(Dettmers et al., 2022; Liu et al., 2024; Guo et al., 2024; Li et al., 2024; Kopiczko et al., 2024; Nikdan et al., 2024; Dettmers et al., 2024). These methods have achieved exceptional performance, often comparable to dense fine-tuning across a range of tasks.

**Neural Network Pruning**: Neural network pruning is a widely applied technique that leverages parameter sparsity to decrease model complexity and accelerate inference (LeCun et al., 1989; Han et al., 2015; Han, 2017; Hoefler et al., 2021). Most pruning methods first evaluate the "importance" of neural network weights (or neurons), and remove the least important parameters. *Unstructured* pruning methods maintain the network architecture (number of layers and the number of neurons within a layer) but zero out a large subset of the weights. *Structured* pruning, on the other hand, finds

---

[3]NVIDIA's memory optimization techniques are available at `https://pytorch.org/torchtune/stable/tutorials/memory_optimizations.html`

the dependency of parameters (Liu et al., 2021; Fang et al., 2023; Ma et al., 2023), evaluates the importance of parameters by group, and removes the dependent group of components like channels or neurons, thereby reducing the network's size. Both of these methods rely on subsequently retraining the model to recover the accuracy that may be lost during pruning. While network pruning has been very successful for classical NN architectures, applying the methods to LLMs can be very expensive: first, computing importance scores by gradient-based methods can require large memory budgets; second, the retraining step can be prohibitive for large models. Thus, memory efficient LLM pruning has been a research area in itself (Sun et al., 2024; Frantar & Alistarh, 2023).

## 3 OUR METHOD

We now present our main contribution, Structured-Pruning-based Sparse Fine-Tuning (SPSFT), illustrated in Figure 1.

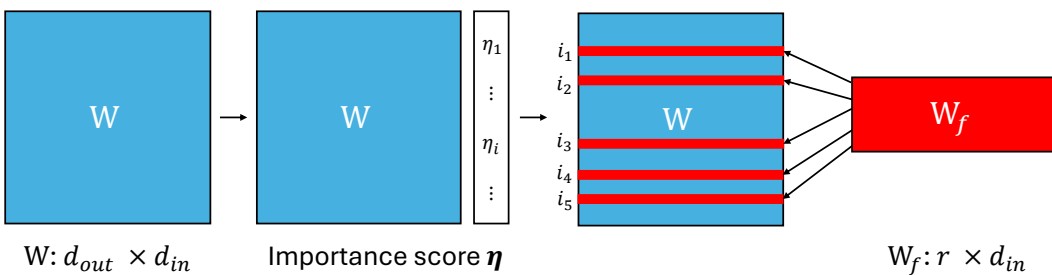

W: $d_{out} \times d_{in}$      Importance score $\boldsymbol{\eta}$      $W_f$: $r \times d_{in}$

Figure 1: The illustration of SPSFT: we evaluate the importance score for each neuron to select the fine-tuning indices. Then we construct the lower-dimensional fine-tuning parameter matrix $\mathbf{W}_f$.

### 3.1 PROPOSED METHOD

SPSFT utilizes structured neural network pruning to select a subset of the parameters for fine-tuning. We evaluate the importance score $\boldsymbol{\eta}$ for each of the neurons and select the ones with the top-$r$ importance scores, where $r$ is a parameter that is decided by the desired number of fine-tuning parameters. The choice of the importance score turns out to be important and we discuss it in detail in Section 3.2. Suppose the indices of the top $r$ neurons are $i_1, i_2, \ldots, i_r$. We next construct a lower-dimensional parameter matrix $\mathbf{W}_f \in \mathbb{R}^{r \times d_{in}}$ and a row selection matrix $\mathbf{M} \in \mathbb{R}^{d_{out} \times r}$ that is zero everywhere, except for $\mathbf{M}_{i_j j} = 1$ for all $j \in [r]$. Following the notations defined in Section 2, we initialize $\mathbf{W}_f$ to be zeros and the final parameters $\hat{\mathbf{W}}$ are defined as Equation 1 where $\mathbf{W}_s = \mathbf{M}\mathbf{W}_f$.

Let us now examine how to implement the backpropagation step so as to be memory-efficient. If the computation graph were to pass through $\mathbf{W} + \mathbf{M}\mathbf{W}_f$ (as a naïve implementation would), the gradients would be computed for all $d_{in} \times d_{out}$ parameters, which is redundant. Instead, we use the additivity of the forward function: we have, analogous to the setting of LoRA,

$$f(\hat{\mathbf{W}}, \mathbf{x}) = f(\mathbf{W} + \mathbf{M}\mathbf{W}_f, \mathbf{x}) = f(\mathbf{W}, \mathbf{x}) + f(\mathbf{M}\mathbf{W}_f, \mathbf{x}), \tag{5}$$

Since $\mathbf{W}$ remains frozen during fine-tuning, backpropagation only needs to keep track of the derivatives of the second term on the RHS. Since $\mathbf{M}$ is now a *fixed* matrix, the only trainable parameters are those in $\mathbf{W}_f$. Therefore, $f(\mathbf{W}_f, \mathbf{x})$ will not be cached, while LoRA requires the cache of $f(\mathbf{A}, \mathbf{x})$ for computing $\frac{\partial \mathbf{h}}{\partial \mathbf{B}}$ (backpropagation, Rumelhart et al. 1986). We explain this in detail in Appendix D.4 and we show that this benefit of memory usage is significant in Section 5.3. We discuss the memory usage and comparison with the LoRA method of (Hu et al., 2022) in Section 5.4.

An important strength of our approach is its flexibility: it can easily incorporate any desired choice of importance scores. At the other end, it can also incorporate new ideas in PEFT research. For example, quantization (QLoRA, Dettmers et al. 2024), parameter sharing (VeRA, Kopiczko et al. 2024), and combining SFT with LoRA (RoSA, Nikdan et al. 2024) can be used.

## 3.2 IMPORTANCE METRIC

Importance evaluation plays a crucial role in our approach, as discussed above. We try various choices in our work: the first is the simple $\ell_2$ norm of the weight vector corresponding to each neuron; the second is the widely-used Taylor importance score (LeCun et al., 1989). We also consider different variants of Taylor importance, as we discuss below. However, for large models like Llama-3, it turns out that the computational overhead required for computing Taylor importances is already prohibitively large![4] In these cases, we only use the norm (or magnitude) of the weight vector per neuron as the importance score. We remark that norm-based importance can be quite powerful on its own, as is the case with norm-sampling in the matrix approximation literature (Frieze et al., 2004).

In our experiments on image classification tasks, we also consider a "class aware" variant of Taylor importance, which may be of independent interest. The motivation here comes from the observation that the importance of a neuron may depend on the class of an input example (as a toy example, a whisker detecting neuron may be very important to the cat class, but not much to others; hence not too important on average). Another motivation comes from the observation that when we perform a vanilla (class agnostic) fine-tuning, the accuracy of some classes can be much worse than others — an undesirable outcome. This is shown in Table 1.

|  | #labels | Mean | Min | Q1 | Med | Q3 | Max |
|---|---|---|---|---|---|---|---|
| CIFAR100 | 100 | 90.18 | 65 | 88 | 92 | 95 | 99 |
| Tiny-ImageNet | 200 | 87.55 | 62 | 84 | 88 | 92 | 100 |

Table 1: The distribution of accuracies across different labels is summarized by statistics including the minimum (Min), first quartile (Q1), median (Med), third quartile (Q3), and maximum (Max) accuracies. #labels is the number of labels. The reported accuracies are the validation results of dense fine-tuned DeiT for 5 epochs. Models and Datasets are described in Section 4.

We define the class-wise Taylor importance as follows: for neuron $i$ and label $t$,

$$\boldsymbol{\eta}_i^t := |L(\mathcal{D}^t, \acute{F}_{\boldsymbol{c}_i}) - L(\mathcal{D}^t, F)| \approx |\mathbf{w}^\top \nabla_{\mathbf{w}} L(\mathcal{D}^t, F)|, \tag{6}$$

where $F$ is the forward function of the entire model, $L(\mathcal{D}^t, F)$ denotes the average loss of $F$ over inputs in class $t$, $\acute{F}_{\boldsymbol{c}_i}$ represents the forward without channel/neuron $\boldsymbol{c}_i$, and $\mathbf{w}$ is the parameter vector of channel $\boldsymbol{c}_i$. One natural choice of importance of neuron $i$ motivated by the above discussion is $\max_t \boldsymbol{\eta}_i^t$. We find that this score is "too sensitive" (importance of neurons may be over-estimated because of just one class), leading to lower overall accuracy. On the other hand, the average (over $t$) of $\eta_i^t$ corresponds to the standard Taylor importance. We find that the intermediate quantity of Quantiles-Mean, defined as the average of the $0\%, 10\%, 20\%, \ldots, 100\%$ quantiles of the $\boldsymbol{\eta}_i^t$, works well in reducing the accuracy imbalance across labels, and also achieving a high overall accuracy. Formally,

$$\boldsymbol{\eta}_i = \frac{\sum_{k=0}^{10} Q_k(\{\boldsymbol{\eta}_i^t\}_{t=1}^p)}{11}, \tag{7}$$

where $Q_k$ represents the $k \times 10$-th quantile. See Appendix A for more details.

## 4 EXPERIMENTAL SETUP

### 4.1 DATASETS

We use several datasets for different tasks. For image classification tasks, we use Tiny-ImageNet (Tavanaei, 2020), CIFAR100 (Krizhevsky et al., 2009), and caltech101(Li et al., 2022) to analyze the fine-tuning strategies. For language classification tasks, we use GLUE (Wang et al., 2019) for training and evaluation. In these tasks, we fine-tune the models on the training split and analyze the results on validation split. For text generation tasks, we will randomly sample 256 instances from the training split of Stanford-Alpaca dataset (Taori et al., 2023) to fine-tune LLM, then evaluate the zero-shot performance on 7 tasks from EleutherAI LM Harness (Gao et al., 2021). More details of datasets can be found in Appendix C.

---

[4]The Taylor importance here refers to computing the exact value without relying on approximations of the importance score or the gradient matrix used for deriving the importance score.

## 4.2 MODELS AND BASELINES

We begin by fine-tuning small models to analyze fine-tuning strategies and select hyperparameters. This includes fine-tuning DeiT (Touvron et al., 2021), ViT, ResNet101 (He et al., 2016), and ResNeXt101 (Xie et al., 2017) on CIFAR100 and Tiny-ImageNet as well as fine-tuning DeBERTaV3-base (He et al., 2023) on GLUE. We compare the results of our SPSFT to fine-tuning the entire model and fine-tuning only the classification layers. For simplicity, we refer to these as dense fine-tuning and head fine-tuning, respectively. Dense fine-tuning serves as the baseline. In these experiments, we fix the fine-tuning ratio at 5% for our approach, meaning the total number of fine-tuning parameters will be approximately 5% of the backbone parameters plus the parameters of the classification layers.

We then fine-tune the full-precision Llama-2-7B and Llama-3-8B, i.e. float32, using our SPSFT and baseline. While DoRA (Liu et al., 2024), RoSA (Nikdan et al., 2024), and some other LoRA variants have shown improvements, they often come at the cost of increased time complexity, memory complexity, or both. Similarly, most SFT methods and dense fine-tuning demand substantial memory during training. Therefore, we use LoRA as the baseline in our experiments. The classification layers of Llama are frozen, and we only fine-tune the linear layers in the attention blocks and the subsequent MLP blocks.

## 4.3 TRAINING DETAILS

We implement our fine-tuning framework based on torch-pruning[5] (Fang et al., 2023), PyTorch (Paszke et al., 2019), PyTorch-Image-Models (Wightman, 2019), and the HuggingFace Transformers library (Wolf et al., 2020). All the experiments are conducted on a single A100-80GB GPU and the optimizer is Adam (Kingma & Ba, 2015). We fine-tune all models for a fixed number of epochs, without performing model selection based on validation data.

Structured pruning often considers parameter dependencies in importance evaluation (Liu et al., 2021; Fang et al., 2023; Ma et al., 2023). This becomes the following process in our work: first, searching for dependencies using a structured pruning tool; next, evaluating the importance of parameter groups; and finally, fine-tuning the parameters within those important groups collectively. For instance, if $\mathbf{W}_{\cdot j}^a$ and $\mathbf{W}_{i\cdot}^b$ are dependent, where $\mathbf{W}_{\cdot j}^a$ is the $j$-th column in parameter matrix of layer $a$ and $\mathbf{W}_{i\cdot}^b$ is the $i$-th row in parameter matrix of layer $b$, then $\mathbf{W}_{\cdot j}^a$ and $\mathbf{W}_{i\cdot}^b$ will be fine-tuned simultaneously while the corresponding $\mathbf{M}_{dep}^a$ for $\mathbf{W}_{\cdot j}^a$ becomes column selection matrix and $\mathbf{W}_s^a$ becomes $\mathbf{W}_{f,dep}^a \mathbf{M}_{dep}^a$. Consequently, 2.5% fine-tuning parameters for layer $b$ will result in additional 2.5% fine-tuning parameters in each dependent layer (e.g. layer $a$ has $\mathbf{W}_f^a$ and $\mathbf{W}_{f,dep}^a$). Therefore, for the 5% of desired fine-tuning ratio, the fine-tuning ratio with considering dependencies is set to 2.5%[6] for the approach that includes dependencies. More details for dependencies of NN can be found in Appendix B.

**Image models**: The learning rate is set to $10^{-4}$ with cosine annealing decay (Loshchilov & Hutter, 2017), where the minimum learning rate is $\eta_{min} = 10^{-9}$. All image models used in this study are pre-trained on ImageNet.

**DeBERTaV3**: The learning rate is set to $2 \cdot 10^{-5}$ with linear decay, where the decay rate is 0.01. The model is fine-tuned on the full training split of 8 tasks from the GLUE benchmark. The maximum sequence length is fixed to 256 with longer sequences truncated and shorter sequences padded.

**Llama**: For LoRA, we fix $\alpha = 16$, with a dropout rate of 0.1. The learning rate is set to $10^{-4}$ with linear decay, and the decay rate is 0.01. For our SPSFT method, we control the trainable parameters by using rank instead of fine-tuning ratio to intuitively compare with LoRA. The learning rate is set to $2 \cdot 10^{-5}$ with the same decay setting. The linear learning rate decays are applied following a warmup phase over the first 3% of training steps. The maximum sequence length is fixed to 2048 with longer sequences truncated and shorter sequences padded.

---

[5]Torch-pruning is not required, all their implementations are based on PyTorch.

[6]In some complex models, considering dependencies results in slightly more than twice the number of trainable parameters. However, in most cases, the factor is 2.

## 5 RESULTS AND DISCUSSION

We now present the results of fine-tuning image and language classification models using our framework. Whenever possible, we report three results: dense-fine-tuning, our method SPSFT, and head fine-tuning (where all the layer weights are frozen and only a "classification head" is added in the end). We show results across several epochs to compare how training evolves for the different fine-tuning strategies. Following this, we examine the performance of our approach by utilizing various importance metrics and evaluating the impact of involving parameter *dependencies*, as we will explain. Finally, we apply our approach SPSFT to fine-tuning Llama models and compare the results with those obtained using LoRA. Note that memory efficiency is not emphasized for small-scale models, as dataset-related memory—particularly with large batch sizes—dominates consumption in these cases. The main advantage of our method in these cases is the reduced FLOPs due to fewer trainable parameters.

### 5.1 HYPERPARAMETER SETTINGS

| | CIFAR100 | | | Tiny-ImageNet | | | Caltech101 | | |
|---|---|---|---|---|---|---|---|---|---|
| | Dense | Head | SPSFT | Dense | Head | SPSFT | Dense | Head | SPSFT |
| #ep | loss, acc | loss, acc | loss, acc | loss, acc | loss, acc | loss, acc | loss, acc | loss, acc | loss, acc |
| | DeiT | | | DeiT | | | DeiT | | |
| #param: | 86.0M | 0.2M | 4.6M | 86.1M | 0.3M | 4.8M | 86.0M | 0.2M | 4.6M |
| 5 | **0.36, 90.18** | 0.76, 80.25 | **0.37**, 88.70 | **0.54, 87.55** | 0.60, 85.09 | **0.40, 89.69** | 0.11, 97.33 | 1.09, 89.02 | 0.30, 95.41 |
| 10 | 0.44, 90.04 | 0.64, 81.83 | 0.42, 88.62 | 0.69, 86.32 | 0.54, 85.72 | 0.49, 88.96 | **0.11, 97.55** | 0.53, 93.22 | 0.17, 96.28 |
| 30 | 0.62, 89.03 | **0.55, 83.42** | 0.64, 88.61 | 0.94, 84.27 | **0.52, 86.06** | 0.72, 88.67 | 0.11, 97.11 | **0.22, 95.06** | **0.12, 96.50** |
| | ViT | | | ViT | | | ViT | | |
| #param: | 85.9M | 0.1M | 4.5M | 86.0M | 0.2M | 4.6M | 85.9M | 0.1M | 45.2M |
| 5 | **0.38, 90.13** | 1.01, 74.78 | **0.40, 88.13** | 0.51, 88.45 | 0.65, 84.10 | **0.36, 90.87** | 0.12, 97.16 | 1.60, 85.70 | 0.43, 93.96 |
| 10 | 0.45, 89.85 | 0.85, 77.05 | 0.45, 87.55 | 0.66, 86.78 | 0.58, 84.95 | 0.44, 90.48 | **0.11**, 97.20 | 0.85, 89.98 | 0.23, 95.54 |
| 30 | 0.62, 88.78 | **0.71, 79.51** | 0.69, 87.83 | 0.96, 84.20 | **0.55, 85.49** | 0.61, 90.56 | 0.12, **97.24** | **0.33, 92.65** | **0.16, 96.02** |
| | ResNet101 | | | ResNet101 | | | ResNet101 | | |
| #param: | 42.7M | 0.2M | 2.2M | 42.9M | 0.4M | 2.4M | 42.7M | 0.2M | 2.2M |
| 5 | **0.50**, 86.21 | 1.62, 60.78 | **0.59**, 82.36 | **0.92, 77.78** | 1.64, 62.06 | **0.76, 79.66** | 0.14, 96.50 | 1.25, 82.33 | 0.48, 92.56 |
| 10 | 0.58, **86.41** | 1.39, 63.06 | 0.60, 82.33 | 1.10, 76.81 | 1.50, 63.19 | 0.79, 79.54 | **0.14, 96.54** | 0.69, 90.24 | 0.23, 95.58 |
| 30 | 0.80, 84.72 | **1.21, 65.63** | 0.80, **82.49** | 1.54, 74.09 | **1.43, 64.47** | 1.08, 78.58 | 0.18, 95.80 | **0.31, 93.00** | **0.16, 95.89** |
| | ResNeXt101 | | | ResNeXt101 | | | ResNeXt101 | | |
| #param: | 87.0M | 0.2M | 4.9M | 87.2M | 0.4M | 5.1M | 87.0M | 0.2M | 4.9M |
| 5 | **0.47, 87.30** | 1.42, 65.07 | **0.47**, 85.94 | **0.86, 79.51** | 1.46, 65.59 | **0.61, 83.88** | **0.12, 97.07** | 1.25, 83.16 | 0.28, 95.84 |
| 10 | 0.56, 87.17 | 1.23, 67.55 | 0.53, 86.04 | 1.01, 79.27 | 1.35, 66.73 | 0.69, 83.47 | 0.13, 96.89 | 0.68, 90.94 | 0.18, 96.28 |
| 30 | 0.71, 86.59 | **1.08, 69.45** | 0.69, **86.33** | 1.41, 76.55 | **1.29, 67.93** | 0.90, 82.83 | 0.16, 96.63 | **0.31, 92.87** | **0.14, 96.76** |

Table 2: Fine-tuning on CIFAR100 and Tiny-ImageNet. #ep and #param represent the number of epochs and the number of trainable parameters, where SPSFT is our method with Taylor importance. All reported losses and accuracies are based on validation results. **Bold** denotes the best results of each fine-tuning approach (in the same column) on the same model and dataset.

We report the results of three approaches over several epochs as table 2 and table 3. Overall, dense fine-tuning over higher epochs is more prone to overfitting, while head fine-tuning shows the exact opposite trend. Except for the results on caltech101[7], the loss patterns across all models consistently reflect this trend, and most accuracy results further support this conclusion. However, our approach demonstrates a crucial advantage by effectively balancing the tradeoff between performance and computational resources.

Table 2 clearly shows that both our approach and dense fine-tuning achieve optimal results within a few epochs, while head fine-tuning requires more training. Notably, all models have been pre-trained on ImageNet-1k, which may explain the strong performance observed with head fine-tuning on Tiny-ImageNet. However, even with this advantage, dense fine-tuning still outperforms head fine-tuning, and our approach surpasses both. In just 5 epochs, our approach achieves results comparable to dense fine-tuning on all datasets with significantly lower trainable parameters.

---

[7]The inconsistent trend observed in Caltech101 results is likely due to its significantly smaller sample size.

| method | #param | task | CoLA | MNLI | MRPC | QNLI | QQP | RTE | SST-2 | STS-B |
|---|---|---|---|---|---|---|---|---|---|---|
| | | #train | 8.5k | 393k | 3.7k | 108k | 364k | 2.5k | 67k | 7k |
| | | epochs | mcc | acc | acc | acc | acc | acc | acc | corr |
| Dense | 184.42M | 3 | **69.96** | **89.42** | 89.71 | **93.57** | 92.08 | 80.14 | **95.53** | 90.44 |
| Dense | | 5 | 69.48 | 89.29 | 87.74 | 93.36 | 92.08 | **83.39** | 94.72 | 90.14 |
| Dense | | 10 | 68.98 | 88.55 | **90.20** | 93.15 | 91.97 | 80.51 | 93.81 | **90.71** |
| Head | 592.13K | 3 | 24.04 | 62.64 | **68.38** | 70.73 | 80.18 | **52.71** | 65.48 | 5.66 |
| Head | | 5 | 45.39 | 61.75 | **68.38** | **72.32** | 80.59 | 47.29 | **78.44** | 26.88 |
| Head | | 10 | **47.32** | **63.98** | **68.38** | 71.99 | **80.96** | 47.29 | 74.66 | **49.59** |
| SPSFT | 103.57M | 3 | 64.08 | 89.58 | 81.62 | 93.10 | 90.70 | 70.40 | 95.18 | 86.58 |
| SPSFT | | 5 | 65.40 | **90.21** | 86.03 | **93.17** | 90.93 | 74.37 | 95.30 | 87.36 |
| SPSFT | | 10 | **65.56** | 89.55 | **87.50** | 93.15 | **91.57** | **80.14** | **95.41** | **89.14** |

Table 3: Fine-tuning DeBERTaV3 on GLUE. 'mcc', 'acc', and 'corr' represent 'Matthews correlation', 'accuracy', and 'Pearson correlation', respectively. #param is the number of trainable parameters. All reported metrics are based on validation results, and are percentages. **Bold** denotes the best results of each fine-tuning approach on the same task.

In contrast to Table 2, the results in Table 3 show more variation. Although the validation loss follows a similar trend, we report only the evaluation metrics due to the different patterns observed in these metrics. One potential reason for this variation is the varying amounts of training data across the GLUE tasks. As shown in the table, tasks with fewer samples often require more epochs to achieve better performance for both dense fine-tuning and our approach. Conversely, for tasks with large amounts of training data such as 'MNLI', 'QNLI', 'QQP', and 'SST-2', the results show tiny improvement from 3 to 10 epochs. Nevertheless, the results still demonstrate that our approach significantly balances the tradeoff between performance and computational resources. Our method achieves near dense fine-tuning performance with remarkably less trainable parameters.

## 5.2 Importance Score and Parameter Dependency

| model | data dep | CIFAR100 | | | Tiny-ImageNet | | | Caltech101 | |
|---|---|---|---|---|---|---|---|---|---|
| | | $\ell^2$ | Taylor | QMTaylor | $\ell^2$ | Taylor | QMTaylor | $\ell^2$ | Taylor |
| DeiT | ✗ | **88.05** | **88.70** | **89.37** | **89.31** | **89.69** | **89.75** | **95.01** | **95.41** |
| | ✓ | 86.43 | 87.33 | 88.08 | 85.56 | 85.92 | 86.49 | 65.35 | 78.04 |
| ViT | ✗ | **87.13** | **88.06** | **88.51** | **90.78** | **90.87** | **90.90** | **92.69** | **93.96** |
| | ✓ | 85.24 | 86.83 | 87.91 | 88.83 | 88.95 | 89.67 | 56.30 | 77.82 |
| RN | ✗ | **82.25** | **82.36** | **83.50** | **79.83** | **79.66** | **80.02** | **93.13** | 92.56 |
| | ✓ | 78.63 | 78.62 | 81.18 | 69.87 | 69.24 | 72.51 | 54.68 | 52.71 |
| RNX | ✗ | **86.12** | **85.94** | **86.93** | **83.88** | **83.88** | **84.17** | **95.71** | **95.84** |
| | ✓ | 84.71 | 85.01 | 85.48 | 79.39 | 78.95 | 79.54 | 92.13 | 91.82 |

Table 4: Fine-tuning image models by our SPSFT for 5 epochs. "dep" refers to whether parameter dependencies are involved or not. $\ell^2$, Taylor, and QMTaylor represent the magnitude, Taylor importance, and Quantiles-Mean Taylor importance (Equation 7). Note that QMTaylor is not applied to fine-tuning Caltech101 due to its significantly imbalanced labels. All reported results are validation accuracies. **Bold** indicates the superior results achieved through dependency searching compared to not searching. Underline highlights the best fine-tuning results.

We utilize various importance metrics to fine-tune both models using our approach, with and without incorporating parameter dependencies, and report the results to compare their performances. Searching for dependencies in structured pruning is natural, as dependent parameters are pruned together. However, important neurons in a given layer do not always have dependent neurons that are also important in their respective layers. As demonstrated in Table 4, fine-tuning without considering parameter dependencies outperforms fine-tuning incorporating dependencies in all cases. For importance metrics, although the differences between them are not substantial, all results consistently conclude that the Quantile-Mean Taylor importance demonstrates a slight improvement over the standard Taylor importance. Furthermore, both the Quantile-Mean Taylor and standard Taylor metrics outperform the magnitude importance.

Table 5 suggests a different conclusion: the impact of parameter dependencies on performance is minor, nearly negligible[8]. However, searching for dependencies involves additional implementations and computational overhead. Combining the results of image models, the conclusion is not searching for the parameter dependencies. For importance metrics, this experiment shows that magnitude and Taylor importance perform similarly.

| imp | task
dep | CoLA
mcc | MNLI
acc | MRPC
acc | QNLI
acc | QQP
acc | RTE
acc | SST-2
acc | STS-B
corr |
|---|---|---|---|---|---|---|---|---|---|
| Taylor | ✗ | 65.56 | 89.55 | **87.50** | 93.15 | **91.57** | **80.14** | **95.41** | 89.14 |
| | ✓ | **67.49** | **89.85** | 87.25 | **93.30** | 91.63 | 79.42 | 95.07 | **89.98** |
| $\ell^2$ | ✗ | 65.40 | 89.77 | 83.33 | 92.64 | 91.34 | 74.73 | 94.04 | **88.69** |
| | ✓ | **66.80** | **90.22** | **84.07** | **93.94** | **91.57** | **79.06** | **95.07** | 87.39 |

Table 5: Fine-tuning DeBERTaV3 on GLUE by our SPSFT for 10 epochs. "dep" refers to whether parameter dependencies are involved or not. Taylor and $\ell^2$ indicate the magnitude and Taylor importance. The importance score is Taylor. We do not apply QMTaylor since the number of labels is tiny. 'mcc', 'acc', and 'corr' represent 'Matthews correlation', 'accuracy', and 'Pearson correlation', respectively. All reported metrics are based on validation results. **Bold** indicates the best results of whether considering dependencies.

## 5.3 MAIN RESULTS OF LLM

We apply our SPSFT method to fine-tuning Llama2-7B and Llama3-8B, comparing the results with those obtained through LoRA fine-tuning. We use the magnitude of the neuron vector as the importance metric due to its lower memory requirements. In contrast, gradient-based metrics such as Taylor and Hessian are as memory-intensive as dense fine-tuning of LLMs. Notably, Sun et al. (2024) propose Wanda, a memory-efficient metric for pruning LLMs. However, it still necessitates one epoch of data forwarding and requires memory more than inference for computing the input vector's norm. For epochs choosing, Table 3 shows that 5 or 10 epochs are reasonable for tasks with less than 10,000 training samples. Given that the maximum sequence lengths of Llama are longer than DeBERTaV3, we have opted for only 5 epochs and report the corresponding results to balance computational resources and performance.

Table 6 highlights the remarkable memory efficiency of our approach[9]. We explore two fine-tuning strategies: *fine-tuning all linear layers* and *fine-tuning only the MLP layers*, with results presented for both. The former requires slightly more memory for the same number of trainable parameters (see Appendix D.5 for details). Since Llama models are pre-trained on extensive datasets, their attention blocks likely already capture key patterns for token interactions. Our results reveal that freezing these attention blocks maintains performance levels nearly equivalent to fine-tuning all layers. Furthermore, the memory advantage of our approach scales proportionally, with potential savings reaching approximately 50 times for Llama-3-405B.

## 5.4 MEMORY EFFICIENCY

We now discuss the memory usage of our approach compared to LoRA (Hu et al., 2022). In fine-tuning using LoRA, the backpropagation process involves storing gradients for $\mathbf{h}$, $\mathbf{B}$, and $\mathbf{A}$, alongside the parameter values and optimizer states (e.g., momentum). Additionally, for computing $\frac{\partial \mathbf{h}}{\partial \mathbf{B}}$, LoRA either caches $f(\mathbf{A}, \mathbf{x})$ during the forward pass or uses activation checkpointing to recompute $f(\mathbf{A}, \mathbf{x})$ in the backward pass (Herrmann et al., 2019; Singh et al., 2022). Our method offers two key memory-savings: (1) Since $\mathbf{M}$ is not trainable, we eliminate the need to cache the hidden state $f(\mathbf{W}_f, \mathbf{x})$, significantly reducing memory usage. The savings scale with the product of the number of fine-tuning layers, batch size, token length, and rank. Further details are provided in Appendix D.4. (2) The dimensions of $\mathbf{W}_f$ in our approach match those of $\mathbf{A}$, substantially reducing the number of trainable

---

[8]The results of using magnitude importance on the RTE task show significant variation, but this is likely due to the small sample size and the hardness of the task, which result in the unstable performances observed in our experiments. Aside from RTE, the results on other tasks are not significantly different.

[9]Also refer to the full table in Appendix D.3, even with $r = 128$, our method's memory usage remains significantly lower than that of LoRA with $r = 16$.

| Model, ft setting | mem | #param | ARC-c | ARC-e | BoolQ | HS | OBQA | rte | WG | Avg |
|---|---|---|---|---|---|---|---|---|---|---|
| Llama2(7B), LoRA, $r = 64$ | 23.46GB | 159.9M(2.37%) | **44.97** | **77.02** | 77.43 | **57.75** | 32.0 | 62.09 | 68.75 | 60.00 |
| Llama2(7B), SPSFT, $r = 128$ | **17.62GB** | 145.8M(2.16%) | 43.60 | 76.26 | **77.77** | 57.16 | **32.6** | **63.54** | **69.30** | 60.03 |
| Llama2(7B), FA-LoRA, $r = 64$ | 17.25GB | 92.8M(1.38%) | **43.77** | **77.57** | 77.74 | **57.45** | 31.0 | **66.06** | 69.06 | **60.38** |
| Llama2(7B), FA-SPSFT, $r = 128$ | **15.21GB** | 78.6M(1.17%) | 43.00 | 76.22 | **77.83** | 57.11 | **31.2** | 63.54 | **69.38** | 59.75 |
| Llama3(8B), LoRA, $r = 64$ | 30.37GB | 167.8M(2.09%) | **53.07** | **81.40** | **82.32** | **60.67** | 34.2 | 69.68 | **73.56** | **64.98** |
| Llama3(8B), SPSFT, $r = 128$ | **24.49GB** | 159.4M(1.98%) | 52.47 | 80.05 | 81.28 | 60.17 | **34.6** | 70.04 | 72.61 | 64.46 |
| Llama3(8B), FA-LoRA, $r = 64$ | 24.55GB | 113.2M(1.41%) | 52.47 | **81.36** | **82.23** | 60.17 | 35.0 | 70.04 | **73.56** | **64.98** |
| Llama3(8B), FA-SPSFT, $r = 128$ | **22.41GB** | 92.3M(1.15%) | 52.13 | 80.05 | 81.35 | **60.20** | 34.2 | 69.31 | 72.85 | 64.30 |

Table 6: Fine-tuning Llama on Alpaca dataset for 5 epochs and evaluating on 7 tasks from EleutherAI LM Harness. "mem" represents the memory usage, with further details provided in Appendix D.1. #param is the number of trainable parameters, where the difference of #param between the two approaches depends on the architecture of Llama, as some layers have $d_{in} \neq d_{out}$. Note that 10 million trainable parameters only account for less than 0.15GB of memory requirement. FA indicates that we freeze attention layers, but not including MLP layers followed by attention blocks. HS, OBQA, and WG represent HellaSwag, OpenBookQA, and WinoGrande datasets. More details of datasets can be found in Appendix C. The ablation study for different $r$ and the comparison with other LoRA variants can be found in Appendix D. All reported results are accuracies on the corresponding tasks. **Bold** indicates the best results of two approaches on the same task.

parameters under the same rank $r$. While the reduction factor may not always be exactly 2, as $d_{in} \neq d_{out}$ in some layers, the memory efficiency is consistently significant.

Concretely, consider a mixed precision training scenario (Micikevicius et al., 2018) where we fine-tune Llama-3-405B using LoRA with $r = 256$ and the Adam optimizer. Assume the pre-trained model parameters are stored as float32, and the LoRA parameters as float16. Each trainable parameter, along with its gradient, requires a total of 4 bytes, while the Adam optimizer's states (master weights, velocities, and momentums) add another 12 bytes per parameter. For LoRA, the trainable parameters consume approximately $405 \times 2\% \times 16 = 129.6$GB of memory. Using our method with the same rank parameter, memory savings are at least $405 \times 1\% \times 12 = 48.6$GB. Besides, Llama-3-405B supports the token length of 32768 and consists of 126 attention blocks, each with 7 linear layers, so the cache of hidden states $f(\mathbf{A}, \mathbf{x})$ is $126 \times 7 \times b \times 32768 \times 256 \times 4$Byte $= 29b$GB where $b$ is the batch size. This calculation excludes LoRA's dropout layer and the cache of $\frac{\partial L}{\partial f(\mathbf{B}, \mathbf{x})}$ (see Appendix D.4 for details), meaning the actual memory savings in practice can be even greater. Even when the rank is doubled to match LoRA's trainable parameter count, our approach maintains significant memory savings by eliminating the cache requirements.

## 6 CONCLUSIONS AND FUTURE WORK

We propose a parameter-efficient fine-tuning (PEFT) framework that integrates various techniques and importance metrics from model compression, sparse fine tuning (SFT), and matrix decomposition research. Using our method, we can fine-tune LLMs and vision transformers using significantly less computation resources than the popular LoRA (Low-Rank Adaptation) technique, while achieving similar accuracy. We also explore the effects of using different importance metrics from model compression. There are several future directions: (1) For importance metrics, while Quantile-Mean Taylor shows slight improvements, these gains are relatively minor compared to the standard Taylor metric in some cases of DeiT and ViT. We may wish to explore better metrics for classification tasks with a large number of labels. (2) Developing memory-efficient importance metrics for LLMs is another future direction. Gradient-based importance metrics, although effective in small-scale models, are constrained by high memory requirements when applied to LLMs. As LLMs continue to expand in size and complexity, exploring memory-efficient importance metrics that deliver comparable performance is essential for further advancements in this field. (3) Our results show that fine-tuning a small number of neurons can significantly improve model performance on downstream tasks. This observation naturally raises the question: do the selected neurons play a distinctive role in specific tasks? This question is related to the explainability of neural networks, which is an extensive area of research. It will be interesting to understand if (and how) individual neurons chosen for fine-tuning contribute to the new task.

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

# A  IMPORTANCE METRICS

**Taylor importance** is the Taylor expansion of the difference between the loss of the model with and without the target neuron:

$$\eta_i = L(\mathcal{D}, \acute{F}_{\mathbf{c}_i}) - L(\mathcal{D}, F)$$

$$\approx -\mathbf{w}^\top \nabla_{\mathbf{w}} L(\mathcal{D}, F) + \frac{1}{2}\mathbf{w}^\top \nabla_{\mathbf{w}}^2 L(\mathcal{D}, F)\mathbf{w}$$

$$\overset{(*)}{\approx} \frac{1}{2}\mathbf{w}^\top \nabla_{\mathbf{w}}^2 L(\mathcal{D}, F)\mathbf{w}$$

$$\overset{(**)}{\approx} \frac{1}{2}(G\mathbf{w})^\top (G\mathbf{w}),$$

where $G = \nabla_{\mathbf{w}} L(\mathcal{D}, F)$. (**) is from the result of Fisher information Rissanen (1996):

$$\nabla_{\mathbf{w}}^2 L(\mathcal{D}, F) \approx \nabla_{\mathbf{w}} L(\mathcal{D}, F)^\top \nabla_{\mathbf{w}} L(\mathcal{D}, F).$$

Note that (*) is from $\nabla_{\mathbf{w}}L(\mathcal{D}, F) \approx 0$, as removing one channel/neuron from a large neural network typically results in only a negligible reduction in loss. To efficiently compute $\boldsymbol{\eta}_i$, the equation can be further derived as:

$$\boldsymbol{\eta}_i \approx (G\mathbf{w})^\top (G\mathbf{w}) = \sum_j (\frac{1}{|\mathcal{D}|} \sum_{\mathbf{x} \in \mathcal{D}} \frac{\partial L(\mathbf{x}, F)}{\partial w_j} w_j)^2 \approx \sum_j |\frac{1}{|\mathcal{D}|} \sum_{\mathbf{x} \in \mathcal{D}} \frac{\partial L(\mathbf{x}, F)}{\partial w_j} w_j|,$$

where $\mathbf{w} = (w_1, \ldots, w_j, \ldots)$.

**Magnitude importance** is the $\ell_2$-norm of the neuron vector computed as $\sqrt{\sum_j w_j^2}$.

## B  PARAMETER DEPENDENCY

Dependencies of parameters between neurons or channels across different layers exist in NNs. These include basic layer connections, residual connections, tensor concatenations, summations, and more, as shown in Figure 2. The black neurons connected by real lines represent the dependent parameters that are in the same group. Pruning any black neurons results in removing the parameters connected by the real lines. Liu et al. (2021) introduced a group pruning method for CNN models that treats residual connections as grouped dependencies, evaluating and pruning related channels within the same group simultaneously. Similarly, Fang et al. (2023) proposed a novel group pruning technique named Torch-Pruning, which considers various types of dependencies and achieves state-of-the-art results. Ma et al. (2023) further applied this procedure to pruning LLMs. Torch-Pruning can be applied to prune a wide range of neural networks, including image transformers, LLMs, CNNs, and more, making it a popular toolkit for neural network pruning.

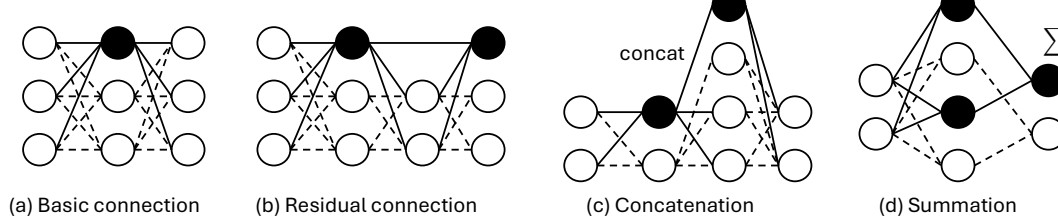

(a) Basic connection     (b) Residual connection     (c) Concatenation     (d) Summation

Figure 2: Common dependencies of parameters in neural networks.

Section 4.3 has described how the parameter dependency works in our approach, we explain it in detail here. Using the same example $\mathbf{W}_{\cdot j}^a$ and $\mathbf{W}_{i\cdot}^b$. Constructing the fine-tuning parameters $\mathbf{W}_f^b$ leads to constructing the fine-tuning parameters $\mathbf{W}_{f,dep}^a$ with the corresponding $\mathbf{M}_{dep}^a$ becoming a column selection matrix and the forward function of layer $a$ becoming the following equation.

$$f(\hat{\mathbf{W}}^a, \mathbf{x}) = f(\mathbf{W}^a, \mathbf{x}) + f(\mathbf{M}^a \mathbf{W}_f^a, \mathbf{x}) + f(\mathbf{W}_{f,dep}^a \mathbf{M}_{dep}^a, \mathbf{x}).$$

Note that in this example, the dependency is connection between the output feature (or channel) of $b$ and the input feature (or channel) of $a$, where $\mathbf{W}^a \in \mathbb{R}^{d_{out}^a \times d_{in}^a}, \mathbf{W}^b \in \mathbb{R}^{d_{out}^b \times d_{in}^b}$ and $d_{in}^a = d_{out}^b$.

## C  DETAILS OF DATASETS

### C.1  VISION BENCHMARKS

**CIFAR100**: CIFAR100 (Krizhevsky et al., 2009) has 100 classes with 600 images of size 32x32 per class, while the CIFAR10 has 10 classes with 6000 images per class. In this study, we use the CIFAR100 downloaded from huggingface (`https://huggingface.co/datasets/uoft-cs/cifar100`) with 500 training images and 100 validation images per class. In our experiments, we resize the images to 256x256, crop the center to 224x224, and normalize them using the CIFAR mean $(0.507, 0.487, 0.441)$ and standard deviation $(0.267, 0.256, 0.276)$ for the three channels.

**Tiny-ImageNet**: Tiny-ImageNet (Tavanaei, 2020) has 200 classes with images of size 64x64, while the full ImageNet-1k (Deng et al., 2009) has all 1000 classes where each image is the standard size 224x224. In this study, we use the Tiny-ImageNet downloaded from huggingface (`https://huggingface.co/datasets/zh-plus/tiny-imagenet`) with 500 training images and 50 validation images per class. In our experiments, we resize the images to 256x256, crop the center to 224x224, and normalize them using the mean $(0.485, 0.456, 0.406)$ and standard deviation $(0.229, 0.224, 0.225)$ for the three channels.

**caltech101**: Caltech101 (Li et al., 2022) consists of 101 classes, with images of varying sizes typically having edge lengths between 200 and 300 pixels. Each class contains approximately 40 to 800 images, resulting in a total of around 9,000 images. In this study, we use the Caltech101 dataset provided by PyTorch (`https://pytorch.org/vision/main/generated/torchvision.datasets.Caltech101.html`), allocating 75% of the images for training and the remaining 25% for validation. In our experiments, we preprocess the images by resizing them to 256×256, cropping the center to 224×224, and normalizing them using the mean $(0.485, 0.456, 0.406)$ and standard deviation $(0.229, 0.224, 0.225)$ for the three channels.

## C.2 GENERAL LANGUAGE UNDERSTANDING EVALUATION BENCHMARK (GLUE)

**CoLA**: The Corpus of Linguistic Acceptability (CoLA) is a dataset for assessing linguistic acceptability (Warstadt et al., 2018). This task is a binary classification for predicting whether a sentence is grammatically acceptable. The dataset is primarily from books and journal articles on linguistic theory.

**MNLI**: The Multi-Genre Natural Language Inference (MultiNLI) is a dataset designed to evaluate a model's ability to perform natural language inference (NLI). The task is to predict whether the premise entails the hypothesis, contradicts the hypothesis, or neither. The data set contains 433k sentence pairs annotated with textual entailment information (Williams et al., 2018).

**MRPC**: The Microsoft Research Paraphrase Corpus (Dolan & Brockett, 2005) is a dataset designed for evaluating paraphrase detection systems. It consists of sentence pairs, with binary labels of whether the two sentences in the pair are equivalent. The data are automatically extracted from online news and labeled by humans.

**QNLI**: The Stanford Question Answering Dataset (SQuAD) is a dataset designed for machine comprehension of text (Rajpurkar et al., 2016). The dataset consists of question-paragraph pairs, where one of the sentences in the paragraph contains the answer to the corresponding question. The paragraphs are from Wikipedia and the questions are written by human annotators.

**QQP**: The Quora Question Pairs (QQP) dataset is a dataset of question pairs (`https://data.quora.com/First-Quora-Dataset-Release-Question-Pairs`). The task is to determine whether two questions are semantically equivalent.

**RTE**: The Recognizing Textual Entailment (RTE) datasets are a series of challenges that evaluate models' ability to determine whether a premise can entail a given hypothesis (Dagan et al., 2006; Bar-Haim et al., 2006; Giampiccolo et al., 2007; Bentivogli et al., 2009). The data are constructed based on the texts from Wikipedia and news. The datasets have been used to evaluate the performance of both traditional language models and the state-of-the-art LLMs.

**SST-2**: The Stanford Sentiment Treebank is a dataset of sentences extracted from movie reviews (Socher et al., 2013). Each sentence is labeled as either positive or negative. The task is to predict whether the sentence is positive or negative.

**STS-B**: The Semantic Textual Similarity Benchmark (STSB) is a dataset with sentence pairs collected from news headlines, video and image captions, and natural language inference data (Cer et al., 2017). The task is to predict the semantic similarity between pairs of sentences. Each pair of sentences is annotated with a similarity score ranging from 0 to 5, where 0 indicates no semantic similarity and 5 indicates semantically equivalent.

## C.3 TEXT-GENERATION DATASETS

**Stanford Alpaca**: Alpaca is an instruction dataset designed for instruction training of pre-trained language models (Taori et al., 2023). It contains 52002 instruction-response pairs generated by OpenAI's text-davinci-003 engine or written by humans. Note that there is only a training split in this dataset. Models fine-tuned on Alpaca are often evaluated by other tasks like "EleutherAI LM Harness".

**ARC**: The AI2 Reasoning Challenge (ARC) dataset consists of grade-school level, multiple-choice science questions (Clark et al., 2018). ARC dataset includes a Challenge Set and an Easy Set. The easy set contains questions that can be answered with straightforward reasoning, while the challenge set requires deeper understanding and more reasoning skills. The ARC-Easy includes 2251 training samples, 570 validation samples, and 2376 test samples and the ARC-Challenge includes 1119 training samples, 299 validation samples, and 1172 test samples.

**BoolQ**: Boolean Questions (BoolQ) is a dataset of yes/no question answering (Clark et al., 2019) and includes 9427 training samples and 3270 validation samples. The dataset is designed to assess models' comprehension and reasoning abilities. Each example contains question, passage, answer, and title.

**HellaSwag**: HellaSwag is a dataset designed to evaluate the models' abilities in generating reasonable contexts (Zellers et al., 2019). It consists of prompts with a short context followed by multiple possible continuations. The goal is to find the correct or most plausible option. The training set, validation set, and test set have 39905 samples, 10042 samples, 10003 samples, respectively.

**OpenBookQA**: OpenBookQA is a question-answering dataset (Mihaylov et al., 2018) comprising 4957 training samples, 500 validation samples, and 500 test samples. It requires reasoning ability and a deeper understanding of common knowledge to answer questions. Each data contains a short passage with multiple possible answers. The dataset emphasizes the integration of world knowledge and reasoning skills, making it a challenging benchmark for natural language processing models. It tests models' abilities to understand and apply factual information effectively to solve problems.

**WinoGrande**: WinoGrande is a dataset of 44k problems for choosing the right option for a given sentence (Sakaguchi et al., 2021). It includes 40,938 samples in the training set, 1,267 in the validation set, and 1,267 in the test set. The dataset is designed to assess models' commonsense reasoning abilities. The examples contain sentences with fill-in-blanks that require the model to select the most appropriate option to complete the sentence. We implement LoRA and DoRA by the Huggingface PEFT[10] library and apply RoSA[11] by its official implementation.

## D ABLATION STUDIES AND RELATED ANALYSIS

In this section, we first discuss the computational resource requirements for fine-tuning. Next, we provide an ablation study on the impact of different rank settings for our approach and LoRA, as shown in Table 8. Figure 3 illustrates the computation and cache requirements during backpropagation. Finally, Table 9 demonstrates the advantages of freezing self-attention blocks to reduce memory usage while maintaining performance, and Table 10 compares the performances with other PEFT methods.

### D.1 MEMORY MEASUREMENT

In this study, we detail the memory measurement methodology employed. The total memory requirements can be categorized into three main components:

$$\text{mem}_{\text{TTL}} = \text{mem}_{\text{M}} + \text{mem}_{\text{FT}} + \text{mem}_{\text{Aux}},$$

where:

1. $\text{mem}_{\text{TTL}}$ is the total memory consumed during training.
2. $\text{mem}_{\text{M}}$ represents the memory consumed by the base model itself.

---

[10]https://huggingface.co/docs/peft/index

[11]https://github.com/IST-DASLab/peft-rosa

3. $\text{mem}_{FT}$ corresponds to the memory required for the fine-tuning parameters and their gradients.

4. $\text{mem}_{Aux}$ accounts for any additional memory usage, including optimizer states, caching, and other intermediate computations.

We yield $\text{mem}_M$ by measuring the memory usage during inference on the training data using the pre-trained model. The combined memory usage of $\text{mem}_{FT}$ and $\text{mem}_{Aux}$ is calculated as the difference between $\text{mem}_{TTL}$ and $\text{mem}_{Model}$. For simplicity, we consistently report $\text{mem}_{FT} + \text{mem}_{Aux}$ as "mem" in all comparisons presented in this study.

| | Llama2(7B) | | | | Llama3(8B) | | | |
|---|---|---|---|---|---|---|---|---|
| FT setting | #param | $\text{mem}_{TTL}$ | $\text{mem}_M$ | mem | #param | $\text{mem}_{TTL}$ | $\text{mem}_M$ | mem |
| LoRA, $r = 64$ | 159.9M(2.37%) | 53.33GB | 29.87GB | 23.46GB | 167.8M(2.09%) | 64.23GB | 33.86GB | 30.37GB |
| RoSA, $r = 32, d = 1.2\%$ | 157.7M(2.34%) | 74.52GB | 29.87GB | 44.65GB | 167.6M(2.09%) | >80GB | 33.86GB | >46.14GB |
| DoRA, $r = 64$ | 161.3M(2.39%) | 74.67GB | 29.87GB | 44.80GB | 169.1M(2.11%) | >80GB | 33.86GB | >46.14GB |
| SPSFT, $r = 128$ | 145.8M(2.16%) | **47.49GB** | 29.87GB | **17.62GB** | 159.4M(1.98%) | **58.35GB** | 33.86GB | **24.49GB** |
| FA-LoRA, $r = 64$ | 92.8M(1.38%) | 47.12GB | 29.87GB | 17.25GB | 113.2M(1.41%) | 58.41GB | 33.86GB | 30.37GB |
| FA-RoSA, $r = 32, d = 1.2\%$ | 98.3M(1.46%) | 68.16GB | 29.87GB | 38.29GB | 124.3M(1.55%) | 76.09GB | 33.86GB | 42.23GB |
| FA-DoRA, $r = 64$ | 93.6M(1.39%) | 60.44GB | 29.87GB | 30.57GB | 114.3M(1.42%) | >80GB | 33.86GB | >46.14GB |
| FA-SPSFT, $r = 128$ | 78.6M(1.17%) | **45.08GB** | 29.87GB | **15.21GB** | 92.3M(1.15%) | **56.27GB** | 33.86GB | **22.41GB** |

Table 7: The requirements of computation resources for fine-tuning. 'mem' traces $\text{mem}_{TTL} - \text{mem}_M$. All fine-tuning parameters are stored in full precision. We also examined the training time and observed that DoRA requires 50% to 100% more time than other methods, while LoRA, RoSA, and our approach need similar training time (differing only by a few seconds). However, due to the influence of various factors on training time and the difficulty of ensuring a fair comparison, we chose not to include these results in our report.

## D.2 RESOURCE REQUIREMENTS

Table 7 presents the resource requirements of various PEFT methods. We compare our approach with LoRA and several of its variants that maintain or surpass LoRA's performance. As shown, our method is the most resource-efficient among these approaches. The subsequent ablation study further demonstrates that our approach achieves performance comparable to LoRA. We exclude comparisons with VeRA (Kopiczko et al., 2024), which proposes freezing a single pair of random low-rank matrices shared across all layers. While VeRA achieves substantial memory savings, its performance often deteriorates.

We note that while our approach offers significant memory efficiency, this benefit is less pronounced in small-scale models, where the primary memory consumption arises from the dataset—especially with large batch sizes. The main advantage of our method in these cases is the reduced FLOPs due to fewer trainable parameters. Therefore, we do not highlight memory efficiency in small-scale model scenarios.

## D.3 RANK SETTINGS

We present an ablation study of rank settings here. Table 8 demonstrates that $r = 16$ is sufficient for LoRA when fine-tuning Llama-2 and Llama-3. In contrast, increasing $r$ for our approach yields slight performance improvements. The most remarkable observation in Table 8 is the exceptional memory efficiency of our approach: even with $r = 128$, the memory usage of our method is significantly lower than that of LoRA with $r = 16$.

## D.4 CACHE BENEFIT

Figure 3 illustrates the computation and cache requirements in backpropagation (Rumelhart et al., 1986). For simplicity, we replace the notation $f(\cdot, \cdot)$ with different $\mathbf{h}$. With the same number of trainable parameters, our approach eliminates the need to cache $\mathbf{h} = \mathbf{W}_f \mathbf{x}$ shown in the figure. While this benefit is negligible under lower rank settings ($r$) or when the number of fine-tuning layers is small, it becomes significant as the model size and rank settings increase. Although the caching requirement for $\mathbf{h}$ can be addressed by recomputing $\mathbf{h} = \mathbf{Ax}$ during backpropagation, this would

| Model, ft setting | mem | #param | ARC-c | ARC-e | BoolQ | HS | OBQA | rte | WG | Avg |
|---|---|---|---|---|---|---|---|---|---|---|
| Llama2(7B), LoRA, $r=16$ | 21.64GB | 40.0M(0.59%) | **44.71** | **76.89** | 77.49 | **57.94** | **32.2** | 60.65 | 68.75 | **59.80** |
| Llama2(7B), SPSFT, $r=32$ | **15.57GB** | 36.4M(0.54%) | 43.00 | 76.43 | **77.80** | 57.06 | 31.4 | **63.18** | **69.14** | 59.72 |
| Llama2(7B), LoRA, $r=32$ | 22.21GB | 80.0M(1.19%) | **44.28** | **76.89** | 77.37 | **57.61** | 32.0 | **64.62** | 69.14 | **60.27** |
| Llama2(7B), SPSFT, $r=64$ | **16.20GB** | 72.9M(1.08%) | 43.26 | 76.30 | **77.83** | 57.13 | **32.2** | 63.18 | **69.22** | 59.87 |
| Llama2(7B), LoRA, $r=64$ | 23.46GB | 159.9M(2.37%) | **44.97** | **77.02** | 77.43 | **57.75** | 32.0 | 62.09 | 68.75 | 60.00 |
| Llama2(7B), SPSFT, $r=128$ | **17.62GB** | 145.8M(2.16%) | 43.60 | 76.26 | **77.77** | 57.16 | **32.6** | **63.54** | **69.30** | **60.03** |
| Llama2(7B), FA-LoRA, $r=16$ | 16.29GB | 23.2M(0.34%) | 44.54 | 77.36 | **77.83** | 57.39 | 30.8 | **67.15** | 68.82 | **60.56** |
| Llama2(7B), FA-SPSFT, $r=32$ | **14.16GB** | 19.7M(0.29%) | 73.17 | 76.30 | 77.55 | 57.14 | **31.2** | 63.18 | **69.38** | 59.70 |
| Llama2(7B), FA-LoRA, $r=32$ | 16.56GB | 46.4M(0.69%) | **44.03** | **77.48** | 77.61 | **57.40** | 30.4 | **65.70** | 68.98 | **60.23** |
| Llama2(7B), FA-SPSFT, $r=64$ | **14.48GB** | 39.3M(0.58%) | 43.17 | 76.26 | **77.65** | 57.17 | **31.4** | 63.18 | **69.22** | 59.72 |
| Llama2(7B), FA-LoRA, $r=64$ | 17.25GB | 92.8M(1.38%) | **43.77** | **77.57** | 77.74 | **57.45** | 31.0 | **66.06** | 69.06 | **60.38** |
| Llama2(7B), FA-SPSFT, $r=128$ | **15.21GB** | 78.6M(1.17%) | 43.00 | 76.22 | **77.83** | 57.11 | **31.2** | 63.54 | **69.38** | 59.75 |
| Llama3(8B), LoRA, $r=16$ | 28.86GB | 41.9M(0.52%) | **53.50** | **81.44** | **82.35** | **60.61** | 34.2 | 69.31 | **73.56** | **65.00** |
| Llama3(8B), SPSFT, $r=32$ | **22.62GB** | 39.8M(0.50%) | 50.26 | 80.09 | 81.10 | 60.21 | **34.4** | **70.40** | 72.93 | 64.20 |
| Llama3(8B), LoRA, $r=32$ | 29.37GB | 83.9M(1.04%) | **53.33** | **81.86** | **82.20** | **60.65** | 34.0 | 68.23 | **73.72** | **64.85** |
| Llama3(8B), SPSFT, $r=64$ | **23.23GB** | 79.7M(0.99%) | 51.96 | 80.01 | 81.31 | 60.18 | **34.6** | **70.04** | 72.85 | 64.42 |
| Llama3(8B), LoRA, $r=64$ | 30.37GB | 167.8M(2.09%) | **53.07** | **81.40** | **82.32** | **60.67** | 34.2 | 69.68 | **73.56** | **64.98** |
| Llama3(8B), SPSFT, $r=128$ | **24.49GB** | 159.4M(1.98%) | 52.47 | 80.05 | 81.28 | 60.17 | **34.6** | 70.04 | 72.61 | 64.46 |
| Llama3(8B), FA-LoRA, $r=16$ | 23.54GB | 28.3M(0.35%) | 51.45 | 81.48 | 82.17 | 60.17 | **34.4** | 68.95 | 73.48 | 64.59 |
| Llama3(8B), FA-SPSFT, $r=32$ | **21.24GB** | 23.1M(0.29%) | 50.26 | 80.09 | 81.19 | **60.22** | 34.2 | **69.68** | 73.01 | 64.09 |
| Llama3(8B), FA-LoRA, $r=32$ | 23.89GB | 56.6M(0.71%) | **52.22** | **81.61** | **82.35** | **60.26** | 35.0 | 69.68 | **73.80** | **64.99** |
| Llama3(8B), FA-SPSFT, $r=64$ | **21.62GB** | 46.1M(0.57%) | 50.26 | 79.97 | 81.22 | 60.20 | 34.2 | **69.68** | 73.01 | 64.07 |
| Llama3(8B), FA-LoRA, $r=64$ | 24.55GB | 113.2M(1.41%) | **52.47** | **81.36** | **82.23** | 60.17 | **35.0** | 70.04 | **73.56** | **64.98** |
| Llama3(8B), FA-SPSFT, $r=128$ | **22.41GB** | 92.3M(1.15%) | 52.13 | 80.05 | 81.35 | **60.20** | 34.2 | 69.31 | 72.85 | 64.30 |

Table 8: Fine-tuning Llama on Alpaca dataset for 5 epochs and evaluating on 7 tasks from EleutherAI LM Harness. 'mem' is the memory usage, see Appendix D.1. #param is the number of trainable parameters, where the difference of #param between the two approaches depends on the architecture of Llama, as some layers have $d_{in} \neq d_{out}$. Note that 10 million trainable parameters only account for less than 0.15GB of memory requirement. FA indicates that we only fine-tune the MLP layers followed by attention blocks. HS, OBQA, and WG represent HellaSwag, OpenBookQA, and WinoGrande datasets. All reported results are accuracies.

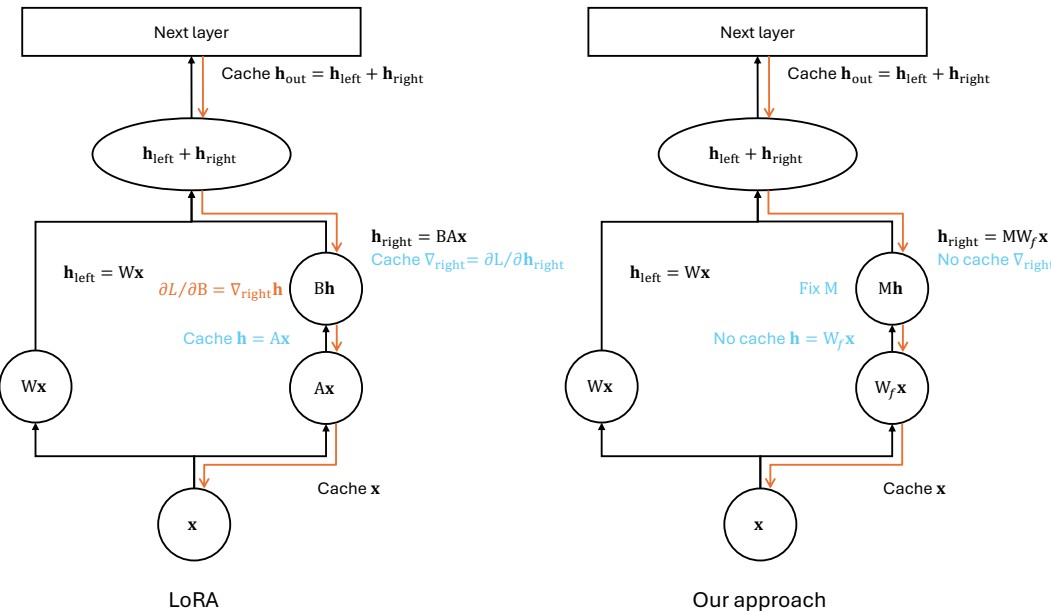

Figure 3: The illustration of backpropagation highlights the operations involved. Black operations occur during the forward pass, while orange operations take place during the backward pass. Blue operations highlight the benefits of our approach. Notably, since $\mathbf{M}$ is non-trainable, caching $\mathbf{W}_f \mathbf{x}$ during the forward pass is unnecessary, leading to significant memory savings. Additionally, in practice, PyTorch caches $\frac{\partial L}{\partial \mathbf{h}_{right}}$ to efficiently compute $\frac{\partial L}{\partial \mathbf{B}}$. This caching is not strictly required for backpropagation.

result in increased time complexity during training. As shown in Table 8, the memory savings range from 1GB to 5GB for various settings of Llama2-7B and Llama3-8B. For much larger models, such as Llama3-405B, this benefit scales proportionally and can be approximately 50 times greater.

We note that additional caches are present in practice. For instance, LoRA includes a dropout layer before computing $\mathbf{Ax}$, which requires extra memory to store the dropout mask. Another example is PyTorch caching $\frac{\partial L}{\partial \mathbf{h}_{\text{right}}}$ to efficiently compute $\frac{\partial L}{\partial \mathbf{B}}$. While this caching is not strictly required, it is employed to optimize implementation and computation efficiency. Consequently, the actual memory-saving advantage of freezing $\mathbf{M}$ is even greater than the reduction in the necessary cache storage, further emphasizing the benefits of our approach.

## D.5 BENEFIT OF FREEZING ATTENTION BLOCKS

| Model, ft setting | mem | #param | ARC-c | ARC-e | BoolQ | HS | OBQA | rte | WG | Avg |
|---|---|---|---|---|---|---|---|---|---|---|
| Llama2(7B), LoRA, $r = 16$ | 21.64GB | 40.0M(0.59%) | **44.71** | 76.89 | 77.49 | **57.94** | **32.2** | 60.65 | 68.75 | 59.80 |
| Llama2(7B), FA-LoRA, $r = 32$ | **16.56GB** | 46.4M(0.69%) | 44.03 | **77.48** | **77.61** | 57.40 | 30.4 | **65.70** | **68.98** | **60.23** |
| Llama2(7B), LoRA, $r = 32$ | 22.21GB | 80.0M(1.19%) | **44.28** | 76.89 | 77.37 | **57.61** | **32.0** | 64.62 | **69.14** | 60.27 |
| Llama2(7B), FA-LoRA, $r = 64$ | **17.25GB** | 92.8M(1.38%) | 43.77 | **77.57** | **77.74** | 57.45 | 31.0 | **66.06** | 69.06 | **60.38** |
| Llama2(7B), SPSFT, $r = 32$ | 15.57GB | 36.4M(0.54%) | 43.00 | **76.43** | **77.80** | 57.06 | 31.4 | **63.18** | 69.14 | **59.72** |
| Llama2(7B), FA-SPSFT, $r = 64$ | **14.48GB** | 39.3M(0.58%) | **43.17** | 76.26 | 77.65 | **57.17** | 31.4 | 63.18 | **69.22** | **59.72** |
| Llama2(7B), SPSFT, $r = 64$ | 16.20GB | 72.9M(1.08%) | **43.26** | **76.30** | 77.83 | **57.13** | **32.2** | 63.18 | 69.22 | **59.87** |
| Llama2(7B), FA-SPSFT, $r = 128$ | **15.21GB** | 78.6M(1.17%) | 43.00 | 76.22 | 77.83 | 57.11 | 31.2 | **63.54** | **69.38** | 59.75 |
| Llama3(8B), LoRA, $r = 16$ | 28.86GB | 41.9M(0.52%) | **53.50** | 81.44 | **82.35** | **60.61** | 34.2 | 69.31 | 73.56 | **65.00** |
| Llama3(8B), FA-LoRA, $r = 32$ | **23.89GB** | 56.6M(0.71%) | 52.22 | **81.61** | **82.35** | 60.26 | **35.0** | **69.68** | **73.80** | 64.99 |
| Llama3(8B), LoRA, $r = 32$ | 29.37GB | 83.9M(1.04%) | **53.33** | **81.86** | 82.20 | **60.65** | 34.0 | 68.23 | **73.72** | **64.85** |
| Llama3(8B), FA-LoRA, $r = 64$ | **24.55GB** | 113.2M(1.41%) | 52.47 | 81.36 | **82.23** | 60.17 | **35.0** | **70.04** | 73.56 | 64.98 |
| Llama3(8B), SPSFT, $r = 32$ | 22.62GB | 39.8M(0.50%) | **50.26** | **80.09** | 81.10 | **60.21** | 34.4 | **70.40** | 72.93 | **64.20** |
| Llama3(8B), FA-SPSFT, $r = 64$ | **21.62GB** | 46.1M(0.57%) | **50.26** | 79.97 | **81.22** | 60.20 | 34.2 | 69.68 | **73.01** | 64.07 |
| Llama3(8B), SPSFT, $r = 64$ | 23.23GB | 79.7M(0.99%) | 51.96 | 80.01 | 81.31 | 60.18 | **34.6** | **70.04** | **72.85** | **64.42** |
| Llama3(8B), FA-SPSFT, $r = 128$ | **22.41GB** | 92.3M(1.15%) | **52.13** | **80.05** | **81.35** | **60.20** | 34.2 | 69.31 | **72.85** | 64.30 |

Table 9: Same results of Table 8. This table is for comparing *fine-tuning all linear layers* with *fine-tuning only the MLP layers*.

We now assess different fine-tuning strategies. Table 9 highlights the importance of selecting fine-tuning layers strategically to minimize redundant memory usage. Freezing the self-attention blocks achieves performance comparable to fine-tuning all layers while significantly reducing memory consumption during training. This efficiency stems from reducing the need to cache intermediate outputs for gradient computation. For example, as illustrated in Figure 3, using LoRA, $\nabla_{out}$ must be cached to compute $\frac{\partial L}{\partial A}$ for the subsequent layer. Freezing the next layer eliminates this caching requirement, further optimizing memory usage.

| Model, ft setting | mem | #param | ARC-c | ARC-e | BoolQ | HS | OBQA | rte | WG | Avg |
|---|---|---|---|---|---|---|---|---|---|---|
| **Llama2(7B)** | | | | | | | | | | |
| LoRA, $r = 64$ | 23.46GB | 159.9M(2.37%) | **44.97** | 77.02 | 77.43 | **57.75** | 32.0 | 62.09 | 68.75 | 60.00 |
| RoSA, $r = 32$, $d = 1.2\%$ | 39.55GB | 157.7M(2.34%) | 43.86 | **77.48** | **77.86** | 57.42 | 32.2 | **63.90** | 69.06 | **60.25** |
| SPSFT, $r = 128$ | **17.62GB** | 145.8M(2.16%) | 43.60 | 76.26 | 77.77 | 57.16 | **32.6** | 63.54 | **69.30** | 60.03 |
| FA-LoRA, $r = 64$ | 17.25GB | 92.8M(1.38%) | 43.77 | **77.57** | 77.74 | **57.45** | 31.0 | **66.06** | 69.06 | **60.38** |
| FA-RoSA, $r = 32$, $d = 1.2\%$ | 35.50GB | 98.3M(1.46%) | **44.28** | 77.02 | 77.68 | 57.22 | 31.0 | 64.26 | 69.22 | 60.10 |
| FA-SPSFT, $r = 128$ | **15.21GB** | 78.6M(1.17%) | 43.00 | 76.22 | **77.83** | 57.11 | **31.2** | 63.54 | **69.38** | 59.75 |
| **Llama3(8B)** | | | | | | | | | | |
| LoRA, $r = 64$ | 30.37GB | 167.8M(2.09%) | **53.07** | **81.40** | **82.32** | 60.67 | 34.2 | 69.68 | **73.56** | **64.98** |
| RoSA, $r = 32$, $d = 1.2\%$ | 42.91GB | 167.6M(2.09%) | 51.28 | 81.27 | 81.80 | 60.18 | 34.4 | 69.31 | 73.16 | 64.49 |
| SPSFT, $r = 128$ | **24.49GB** | 159.4M(1.98%) | 52.47 | 80.05 | 81.28 | 60.17 | **34.6** | **70.04** | 72.61 | 64.46 |
| FA-LoRA, $r = 64$ | 24.55GB | 113.2M(1.41%) | **52.47** | **81.36** | **82.23** | 60.17 | **35.0** | **70.04** | **73.56** | **64.98** |
| FA-RoSA, $r = 32$, $d = 1.2\%$ | 39.01GB | 124.3M(1.55%) | 52.22 | 81.19 | 82.05 | 60.11 | 34.4 | 69.31 | 73.16 | 64.63 |
| FA-SPSFT, $r = 128$ | **22.41GB** | 92.3M(1.15%) | 52.13 | 80.05 | 81.35 | **60.20** | 34.2 | 69.31 | 72.85 | 64.30 |

Table 10: Comparison with RoSA. While we use full precision for our approach and LoRA, we apply mixed precision training with bfloat16 for RoSA's parameters due to memory limitations. Note that the performance of RoSA with full precision on Llama2, Llama2-FA, and Llama3-FA is similar to its performance with mixed precision training.

## D.6 Additional Comparisons

The results in Table 10 compare our approach with other PEFT methods. While the accuracies of these approaches are similar, there are significant differences in memory efficiency. Our approach consistently achieves the best memory savings, demonstrating its advantage in resource-constrained fine-tuning scenarios.

