# OpenReview forum: "Memory-Efficient Fine-Tuning via Structured Neural Network Pruning"
_ICLR.cc/2025/Conference — Submitted to ICLR 2025_

### Official Review · Reviewer_3hdW · 2024-10-29

**Soundness:** 3
**Presentation:** 2
**Contribution:** 2
**Rating:** 5
**Confidence:** 4

**Summary:**

The paper proposes an improvement to the the efficiency of parameter-efficient finetuning (PEFT) by using a finetuning neuron-level mask, rather than a low-rank decomposition, to perform finetuning. This allows the authors to finetune only a small number of parameters, freezing the rest. The fact that the parameters are organized in matrix rows (corresponding to entire neurons) makes the process efficient, and more so than LoRA for the same number of parameters.

**Strengths:**

The paper clearly lays out its method and the arguments for its method and the method itself is overall well-explained. The authors discuss their experimental choices and add an additional, helpful section on the practical considerations of memory requirements when using the Adam optimizer.

The idea of a class-aware Taylor importance is nice and is a useful side contribution of the paper.

**Weaknesses:**

1. This paper unfortunately does not position itself appropriately with regard to the state of the field. In particular, RoSA [1] considers using a sparse finetuning mask combined with a low-rank adaptation, finding that sharing the parameter budget between the two approaches works better than either in isolation.

    Other relevant work is missing as well. For example, model compression methods are also used for model adaptation even without finetuning [2].  The related work section also fully omits any mention of sparse training methods, of which there are many; most relevant to this work might be the Movement Pruning method [3] and the FISH-Mask method [4].

    Overall, I would suggest removing the claim “ours is the first work that can integrate the literature on model compression for morel fine-tuning”. Further, the proposed method should be benchmarked against RoSA, or, better yet, a combination of this paper’s pruning approach with a low-rank component.

1. The claim that “the computational overhead required for computing Taylor importances is already prohibitively large!” (line 187) is not correct. The SparseGPT method, which the authors cite, relies on an approximate Taylor importance. So does the ZipLM method, which applies to structured pruning.

1. For the image classification task, assuming that the models were pretrained on ImageNet, the transfer tasks are very similar. It would have been more informative to also see cases where the downstream tasks are from a quite different distribution - for instance, some of the ones used in [5], such as Aircraft, flowers, etc.

1. The section on considering parameter dependencies in importance evaluation (line 262-271) is not clearly written, and the computation of the parameter budget was confusing. The whole idea of parameter dependencies doesn’t seem to add much, since the approach is actually unhelpful (per the experimental results). I am not sure why the authors make the claim “structured pruning considers parameter dependencies in its importance evaluation” (line 262), and there is no citation given.

1. Some experimental details are lacking. For the image classification tasks, the authors omit any mention of what dataset the models are pretrained on, if any. Additionally, the results do not have any sort of error bars, making it difficult to judge whether the improvements have any significance.

References:

[1] Nikdan, Mahdi et al. “RoSA: Accurate Parameter-Efficient Fine-Tuning via Robust Adaptation.” ArXiv abs/2401.04679 (2024)

[2] Mallya, Arun et al. “Piggyback: Adapting a Single Network to Multiple Tasks by Learning to Mask Weights.” European Conference on Computer Vision (2018).

[3] Sanh, Victor et al. “Movement Pruning: Adaptive Sparsity by Fine-Tuning.” ArXiv abs/2005.07683 (2020)

[4] Sung, Yi-Lin et al. “Training Neural Networks with Fixed Sparse Masks.” ArXiv abs/2111.09839 (2021)

[5] Kornblith, Simon et al. “Do Better ImageNet Models Transfer Better?” 2019 IEEE/CVF Conference on Computer Vision and Pattern Recognition (CVPR) (2018)

**Questions:**

How does the proposed method compare with RoSA? Can even better results be achieved from combining this method with a low-rank component?

Please expand the related work section as outlined in the 'weaknesses', above

Please expand the performance evaluation in computer vision to more OOD tasks as in [5], above

---

> ### Author Response · Authors · 2024-11-26
>
> Thanks for the valuable review!
> 1. We had included sparse fine-tuning (Lines 94–107), FISH-Mask (Line 104), and NN pruning (Lines 122–134) in the initial version. In the revision, we expanded the related work and introduction on the techniques of efficiently training with sparse matrices and we removed the description of “ours is the first work……”.\
> For RoSA, we tested its performance under various settings but did not observe a significant advantage over LoRA. A comparison with RoSA (using mixed precision training with bfloat16 for RoSA) is included in Table 10 in Appendix D.6 of the revised paper. While we include a comparison with RoSA in the paper, we believe that this would not alter our main conclusion. \
> The key insight we aim to emphasize is that replacing unstructured sparse matrices with sparse ones using our method results in more memory-efficient fine-tuning than the popular LoRA, without requiring additional implementations for sparse matrices. We have revised the paper to underscore this point more explicitly.\
> We tested RoSA with $r=32$ and $d=1.2$% in full precision, following the settings suggested in the RoSA paper, and compared it to LoRA with r=64 under our fine-tuning process. Across seven evaluation datasets, RoSA's accuracies were similar to LoRA's—slightly better on some tasks and slightly lower on others. However, RoSA’s memory usage was significantly higher than LoRA’s under these settings.\
> In a separate experiment, we attempted to fine-tune all linear layers of Llama-3-8B using RoSA but encountered out-of-memory (OOM) errors, requiring over 80GB of memory. Notably, under these settings, LoRA’s trainable parameters exceeded RoSA’s by 1M, yet LoRA’s peak memory usage during training was only 64.22GB. In another test with attention-frozen fine-tuning, RoSA's peak memory usage was 76.09GB, compared to LoRA’s 58.41GB. While RoSA's trainable parameters exceeded LoRA's by approximately 11M, this difference would contribute less than 0.2GB to LoRA's memory usage, highlighting a memory inefficiency of RoSA.\
> RoSA relies on custom C++ implementations for its sparse matrix operations, complicating its application and making memory tracing difficult. The latest version of ‘spops’, released by the RoSA authors, failed to install on our device. We had to rely on an older version of ‘spops’ to execute RoSA. This discrepancy in software versions might partly explain the inconsistent conclusions presented in the RoSA paper regarding memory usage.\
> Thanks for bringing the idea of ‘a combination of this paper’s pruning approach with a low-rank component’. We will leave the study of compatibility with other LoRA variants to be a future work, such as RoSA[1], QLoRA[2], and VeRA[3].
> 2. We added a footnote to the revision: “The Taylor importance here refers to computing the exact value without relying on approximations of the importance score or the gradient matrix used for deriving the importance score.”
> 3. Incorporating feedback from reviewer H135, we have included the results on Caltech101 in the revised version. This dataset is also an out-of-distribution (OOD) benchmark referenced in this paper [4].
> 4. Parameter dependency was commonly used in structured pruning (with some citation at line 127), this was the reason we mentioned this paragraph. We cited some papers of structured pruning at section 4.3 again in revision. \
> Maybe there is some part unclear, $W_{\cdot j}^a$ represents the $j$-th input feature of layer $a$, while $W_{i \cdot}^b$ represents the $i$-th output feature of layer $b$, where the two are connected. In structured pruning, these features, $W^a_{\cdot j}$ and $W^b_{i \cdot}$, are pruned collectively. Similarly, in our approach, this dependency becomes fine-tuning these features together. For a visual representation of this relationship, please refer to Figure 2 in Appendix B.
> 5. In the revised version, we have mentioned the datasets on which the models are pre-trained. Additionally, we have reworded the paper to emphasize that "Without requiring additional implementations for sparse matrix operations, our novel SPF framework achieves memory efficiency while maintaining accuracies comparable to the popular LoRA."
>
> [1] Nikdan, Mahdi et al. “RoSA: Accurate Parameter-Efficient Fine-Tuning via Robust Adaptation.” ArXiv abs/2401.04679 (2024) \
> [2] Dettmers, Tim, et al. "Qlora: Efficient finetuning of quantized llms." Advances in Neural Information Processing Systems 36 (2024).\
> [3] Kopiczko, Dawid Jan, Tijmen Blankevoort, and Yuki Markus Asano. "Vera: Vector-based random matrix adaptation." arXiv preprint arXiv:2310.11454 (2023).\
> [4] Kornblith, Simon et al. “Do Better ImageNet Models Transfer Better?” 2019 IEEE/CVF Conference on Computer Vision and Pattern Recognition (CVPR) (2018)

---

> > ### Comment · Reviewer_3hdW · 2024-12-02
> > **thank you for the rebuttal**
> >
> > I thank the authors for the rebuttal and additional experiments. It is unfortunate that, given the short time in the rebuttal phase it was not possible to run image classification transfer learning experiments on the full suite of downstream datasets from [Kornblith, Simon et al. “Do Better ImageNet Models Transfer Better?” ], which would have been far better; in particular, it would still have been interesting to see results on specialized/finegrained datasets, such as Aircraft or Flowers. Likewise, it is unfortunate that the authors encountered installation instructions with RoSA, possibly creating an unfair comparison in terms of memory consumption, and possibly (though unlikely) also accuracy.
> >
> > Given these issues, I think that, while the idea is promising, the paper is unfortunately not ready to be published. Accordingly, I am raising my score to 5, but cannot raise it higher.

---

> > > ### Author Response · Authors · 2024-12-02
> > >
> > > Thank you for your thoughtful response. We agree that the comparison with RoSA may not be entirely fair in terms of memory consumption, as RoSA relies on its custom sparse backpropagation library, spops. However, even if we were to successfully install the latest version of spops, creating a truly fair comparison would remain challenging due to RoSA’s reliance on components beyond PyTorch’s native capabilities, which directly influence its memory footprint.
> > >
> > > We would also like to reemphasize that our primary goal is to introduce a novel sparse fine-tuning framework that achieves **remarkable memory efficiency and comparable accuracy**, all without requiring additional implementations or external dependencies. This emphasis on simplicity and practicality underscores the accessibility and versatility of our approach for a wide range of users and applications.

---

### Official Review · Reviewer_nTJT · 2024-10-30

**Soundness:** 2
**Presentation:** 2
**Contribution:** 3
**Rating:** 5
**Confidence:** 4

**Summary:**

The paper proposes a Structured-Pruning-based Sparse Fine-Tuning (SPSFT) method that combines neural network compression and matrix decomposition techniques to significantly reduce memory and parameter requirements while achieving performance comparable to existing methods such as LoRA in both vision and language tasks. It is the first to apply network pruning to fine-tune large models, particularly by using structured pruning to identify and select important neurons for fine-tuning, achieving parameter and memory efficiency. The paper also provides guidance on hyperparameter and training configuration choices to help users achieve efficient and high-accuracy fine-tuning results across different tasks.

**Strengths:**

(1): The SPSFT method uses structured pruning and matrix decomposition to greatly reduce memory and parameter requirements while maintaining performance, suitable for scenarios with limited computational resources.
(2): The approach is modular, allowing different pruning strategies to be used, making it adaptable to various task requirements.
(3): The method performs well across both vision and language tasks, demonstrating good generalizability.

**Weaknesses:**

(1): Please clarify the term "QTaylor" mentioned in line 365. Ensure it is defined within the context of your study or provide relevant references. Consistent use and clear definition across the manuscript are crucial for reader comprehension.
(2): Could you elaborate on your choice of datasets for the classification tasks? Additionally, please discuss the potential applicability or comparability of your findings to larger, established datasets like ImageNet to enhance the generalizability of your results.
(3) Consider incorporating the standard 8 zero-shot datasets commonly used in fine-tuning tasks on Llama, if feasible. Additionally, including an AVG (average) metric could provide a holistic view of the model’s performance across tasks. If these datasets were not used, please provide a rationale for your selection.
(4): Please provide a detailed comparison of your model's parameter count and performance with LoRA, particularly at comparable scales. Discuss any trade-offs involved, which would help elucidate the practical implications of your approach.
(5):There are several LoRA variants that have reported improvements in parameter efficiency. Could you compare your approach against key variants such as [DoRA,  VeRA, SVFT, LoRA-XS]? This comparison would be valuable in highlighting the distinct advantages or potential areas for enhancement in your method.
(6): The figure presented in the main text could be enhanced for better clarity and information delivery. Consider adding detailed legends, clearer labels, or supplementary figures that demonstrate performance comparisons or efficiency gains. Visual aids are instrumental in conveying complex information effectively.

**Questions:**

(1): Carefully check for spelling errors in the paper.
(2): Provide a fair comparison with LoRA at the same parameter scale, and report performance on 8 datasets along with the AVG metric.
(3): Consider comparing with some of the LoRA variants.

---

> ### Author Response · Authors · 2024-11-26
>
> Thanks for the thorough review! We discuss the questions/comments raised in the “Weakness” section.
> 1. Thanks for pointing this out, we added clarification and did a thorough review in revision.
> 2. We selected these datasets because they are the standard ones used to benchmark fine-tuning. We also ensured that the size of datasets is small enough to train within a day on a single A100. For vision tasks, the main benefit of our approach is using significantly less trainable parameters, i.e. less FLOPs, to achieve the comparable results to dense fine-tuning. (For small-scale models, the models themselves are quite small, and most of the memory footprint comes from the datasets.) These image models have been pre-trained on ImageNet-1k, and the “pre-train then fine-tune” paradigm is often applied when the computation resources are limited. Given these facts, we didn’t fine-tune the models on ImageNet-1k.
> 3. We will try to add this to the final version. We were not able to do this within the rebuttal period given the timeline due to the limited computation resources. However, we've incorporated 7 commonly used datasets in the evaluation of Llama. ‘BoolQ’, ‘HellaSwag’, ‘WinoGrande’, ‘ARC-e’, ‘ARC-c’, and ‘OpenbookQA’ are included in the standard 8 zero-shot datasets, while ‘rte’ is also a challenging dataset. The experimental results of existing works, e.g. DoRA [1], show that Llama performs well on ‘PIQA’ and ‘SIQA’ by fine-tuning with most PEFT methods. So we exclude the two dataset and select other challenging datasets due to the limitation of computation resources.
> 4. In revision, we compared the results of our approach with rank 128 to the result of LoRA with rank 64, where the number of trainable parameters were similar. We  also added the memory footprint for Llama and provided the exact number of trainable parameters but did not emphasize memory efficiency in small-scale model scenarios. In addition, we added the ablation study for different rank settings in Appendix D.
> 5. Maybe there was some part unclear in the initial version, we aimed to clarify that sparse fine-tuning can achieve even lower memory usage than the popular LoRA under comparable settings of trainable parameters by replacing unstructured sparse matrices with structured sparse ones using our approach. This is accomplished without requiring any complex implementations of forward passes, backward passes, or backpropagation, which often rely on custom C++ implementations to enable efficient sparse tensor operations. Consequently, we selected LoRA, the most widely used method, as our baseline for comparison. \
> In revision, we attempted to include results for DoRA, a method that appears to be more widely adopted and validated than other LoRA variants. However, we encountered significant computational challenges. We included the memory usage of DoRA and RoSA[2] (see reviewer 3hdW) in Appendix D.2. Briefly, DoRA’s memory usage and fine-tuning time were substantially higher than those of LoRA. For example, in our experiments with Llama-3, DoRA consistently required over 80GB of memory for training and exhibited training times approximately 50%-100% longer than LoRA. Given our focus on highlighting the computational efficiency of our approach, we have opted to limit our comparisons to LoRA.
> 6. We added Figure 3 in appendix D4. We re-wrote section 5.4 and added  appendix D to describe more on the memory benefit of our approach in detail.
>
> [1] Liu, Shih-Yang, et al. "Dora: Weight-decomposed low-rank adaptation." arXiv preprint arXiv:2402.09353 (2024). \
> [2] Mahdi Nikdan, Soroush Tabesh, Elvir Crn ˇcevi ´c, and Dan Alistarh. RoSA: Accurate parameter-
> efficient fine-tuning via robust adaptation. In Forty-first International Conference on Machine
> Learning, 2024. URL https://openreview.net/forum?id=FYvpxyS43U

---

> > ### Comment · Reviewer_nTJT · 2024-11-27
> >
> > After reviewing the author's response, some of my concerns have been addressed and corrected, so I am willing to increase the score by two points.

---

> > > ### Author Response · Authors · 2024-12-02
> > >
> > > Thank you for your further response and feedback!

---

### Official Review · Reviewer_5q3j · 2024-10-31

**Soundness:** 2
**Presentation:** 2
**Contribution:** 2
**Rating:** 5
**Confidence:** 4

**Summary:**

The paper presented a parameter-efficient fine-tuning method inspired by techniques used in network pruning. The proposed method identifies and fine-tunes only the most important neurons selected using a pre-defined importance metric. The method is similar to low-rank adaptation (LoRA) in that it also formulates the weight updates as the product of two low-rank matrices. However, one of the low-rank matrices is fixed to binary values determined from the importance metric, while the other matrix is initialised as zeros and trained. Therefore, the method achieves high parameter efficiency, and the experiments also demonstrate competitive performance over vision and language models.

**Strengths:**

1. The method is well-grounded in network pruning research, which has long sought to reduce model complexity while retaining performance by focusing on the more important weights of a model. By leveraging structured pruning and importance metrics, i.e., the Taylor score and L2 norm, the proposed method identifies the output channels that contribute most to the performance. The method is thus very well motivated.

2. PEFT methods such as LoRAs already are already parameter efficient. The proposed method further reduces the number of learnable parameters by fixing one of the low-rank matrices. This leads to even lower learnable parameter count and additional memory save.

3. The experiments are fairly comprehensive, covering both vision models, e.g., ViT, ResNet, and language models, e.g., DeBERTa, Llama. These experiments provide ample evidence on the model's efficiency and performance across a wide range of tasks.

**Weaknesses:**

1. There is a lack of strong evidence on the advantage of the proposed method. The title of the paper focuses on the memory efficiency of the method. However, in most of the tables, the memory consumption of the proposed method is not included. There is only a small section (Sec. 5.4) that briefly mentions the memory footprint. This only provides one data point, and is not comprehensive enough to justify the title. More specifically, Table 6 shows a comparison against LoRAs using Llama models, without any conclusive evidence. The proposed method leads to lower parameter count, but also significantly lower performance in many cases. It is perhaps a more fair comparison to either scale both methods to the same parameter count and compare the performance, or the other way around. In addition to this, showing a curve of performance vs. parameter count/memory consumption of the two methods would largely benefit the argument.

2. A few more minor comments.
(a) Tables and figures are placed at the top of the page as a convention.
(b) Some notations are sloppy. In L161, there is no reason to enumerate the values of rows for matrix $M$. Simply use i as the index. In Eq.6, $F - c_i$ is a very crude notation. It's better to just use a new symbol.

**Questions:**

N/A

---

> ### Author Response · Authors · 2024-11-26
>
> We thank the reviewer for the detailed comments and clarifications.
>
> 1.1. We added the memory footprint in tables of Llama (please see the revision). For image models and debertav3, we do not emphasize the memory footprint because the models themselves are quite small, and most of the memory footprint comes from the datasets. The main benefit of our approach is using significantly less trainable parameters, i.e. less FLOPs, to achieve the comparable results to dense fine-tuning. \
> 1.2. For Llama, we compared the results of our approach with those of LoRA’s with (approximately) the same number of trainable parameters. (The “approximate” is because the rank parameter is chosen as a power of 2 for convenience.) We will highlight the fact that “even then the trainable parameters of our approach is 3 times that of LoRA, the memory usage of our method is smaller than that of LoRA.” This is because the memory usage consists of (a) the “base” memory needed to hold the model, (b) the parameters to be trained, (c) auxiliary memory for storing gradients, etc., (d) memory corresponding to intermediate layer activations. Even with (a) and (b) being the same and using the same optimizer, we see our method outperforming LoRA by a significant amount from (c) and (d).\
> 1.3. “Curve of performance vs. parameter count/memory consumption”: thanks for the suggestion – we add this to the revision. We add Appendix D for the detailed study of memory footprints and rank settings.   \
> 2.(a) While we made efforts to position all tables and figures at the top of the pages in the revised version, some had to be placed differently to optimize the use of space in the main text. \
> 2.(b) Regarding the matrix $\boldsymbol{M}$, this is to make our notation consistent with other sparse fine-tuning and LoRA papers. In Eq.6, we replace the notation $F-\boldsymbol{c}_i$, see the revision.

---

> > ### Comment · Reviewer_5q3j · 2024-12-02
> >
> > Thank you for the response.
> >
> > The added memory footprint in Table 6 is very informative for the method, and I encourage the authors to keep it in the revision. Nevertheless, despite the lower memory consumption, the proposed method still trails behind LoRA in performance, which somewhat limits the impact of the work. In addition, the paper could perhaps benefit from some more insights on the formulation of the method. As I understand, it seems that $MW_f$ essentially returns a matrix where the rows are zeros when the corresponding rows in $M$ are zeros, and a simple average across columns of $W_f$ otherwise. It makes sense that since $M$ is selected based on some the norm of the weight matrix, the rows zeroed out are those with small norms in the first place. If this is in fact true, I think it should be added in the paper to help the readers better understand the method. Beside this, it would be more impactful if the paper can demonstrate some theoretical result on how the well proposed formulation can approximate the true weight update, i.e., how does the error caused by this handcrafted $M$ impact the model.

---

> > > ### Author Response · Authors · 2024-12-02
> > >
> > > Thank you for your response and for providing further suggestions! We truly appreciate your time and valuable feedback.

---

### Official Review · Reviewer_H135 · 2024-11-02

**Soundness:** 2
**Presentation:** 2
**Contribution:** 1
**Rating:** 3
**Confidence:** 3

**Summary:**

This paper discusses a method where they start by identifying "important" neurons or nodes using structural pruning metrics and then fine-tune by focusing only on the weights associated with these neurons. The proposed method performs well on the datasets selected by the authors. However, the paper appears to have been completed rashly, with significant issues in the coherence between sections, a lack of basic comparisons with other baseline methods, and missing experiments on more datasets. I believe the paper requires substantial revision to be considered for acceptance. Although I am inclined against accepting it, I am open to changing my view if persuaded.

**Strengths:**

1. The article validates the proposed method with extensive experiments on both vision and language tasks, which helps in establishing the robustness and applicability of the technique across different domains.

2. It provides a comprehensive comparison with existing state-of-the-art methods like LoRA, demonstrating improvements in memory efficiency methods and maintaining comparable accuracy, which is crucial for practical applications.

**Weaknesses:**

1. The layout of tables in the paper has issues. For instance, there is a large gap between Table 2 and Table 3. Additionally, the full names of "dft" and "hft" are not provided in Table 2, and placing your method in the middle of the table makes feels confusing.
2. Lack of experiments on ImageNet-10/CUB200/Caltech101. At least a dataset with high resolution. Since your experiments are mainly conducted on the CIFAR-100 or TinyImageNet, I have a concern that your method only work on the these datasets.
3. Lack of comparisons with existing baseline methods. e.g. LoRAPrune[1].
4. Limited novelty. In light of the extensive discussions around LoRA and its variants, the contribution of these hybrid methods seems insufficient.

[1] Zhang, Mingyang, et al. "Loraprune: Pruning meets low-rank parameter-efficient fine-tuning." arXiv preprint arXiv:2305.18403 (2023).

**Questions:**

Please convince me and make me believe that the weaknesses I identified in the paper do not actually exist.

---

> ### Author Response · Authors · 2024-11-26
>
> Thanks for the comments and suggestions.
>
> 1. (Layout, abbreviations): We will make these changes; the spacing was an unfortunate typesetting issue.
> 2. (More extensive experiments): Following the suggestion, we have done experiments on Caltech-101 and included the results in the updated draft. Notice that while our experiments on image datasets are somewhat limited, we have done much more extensive tests on language datasets. Given that many applications focus on fine-tuning language models like LLaMA, we focused more on them.
> 3. Comparison with LoRA-Prune: note that this is not actually a paper on fine-tuning; instead, their goal is a better compression procedure for large models. That said, there are other methods such as RoSA (see also the response to Reviewer 3hdW), that we have recently compared to. (In short, our procedure maintains accuracy, while not requiring any sparse matrix libraries, and using a considerably smaller amount of memory.)
> 4. Novelty: Maybe there is some part unclear, we aim to improve the memory challenge of sparse fine-tuning (SFT), which is a parallel line of LoRA and its variants. The key challenge of many SFT is that the unstructured sparse matrix requires additional implementation for forward pass, backward pass, and backpropagation computation (e.g. [1], [2], [3], [4], [5]). This often involves optimizing tensor computations by selectively processing only non-zero elements, e.g. torch.sparse [3], compressed sparse column/row (CSC/CSR) [4], semi-structured formats [5] The tradeoff in these approaches lies in the fact that they all increase time complexity to achieve reduced memory complexity. Therefore, some approaches also leverage C++ for acceleration, as seen in works like  [1], [2]. (see their implementation: [6]) This complicates the practical application of these methods and increases the difficulty of further advancing this field. Our method obtains a memory- and parameter- efficiency without any additional implementations of sparse operations in forward pass, backward pass and back-propagation. We added paragraphs in section1 (line 60-68, 87-89) and section 2 (line 123-137) to more clarify the challenge of sparse fine-tuning. We also added some words at the research questions (line 69-72).\
> Additionally, LoRA and its variants leverage matrix decomposition techniques, which are not directly applicable to vector parameters such as LayerNorm and BatchNorm. In contrast, our approach identifies a subset of parameters for fine-tuning by focusing on the most important neurons or channels, making it compatible with vector parameters. We believe this flexibility is another key strength of our work.\
> By comparing our method with LoRA, we aim to demonstrate that structured sparse fine-tuning is a straightforward and memory-efficient solution. It relies solely on standard tensor operations available in regular libraries, all while maintaining competitive performance.
>
> [1] Mahdi Nikdan, Tommaso Pegolotti, Eugenia Iofinova, Eldar Kurtic, and Dan Alistarh. SparseProp:
> Efficient sparse backpropagation for faster training of neural networks at the edge. In Andreas
> Krause, Emma Brunskill, Kyunghyun Cho, Barbara Engelhardt, Sivan Sabato, and Jonathan
> Scarlett (eds.), Proceedings of the 40th International Conference on Machine Learning, volume
> 202 of Proceedings of Machine Learning Research, pp. 26215–26227. PMLR, 23–29 Jul 2023.
> URL https://proceedings.mlr.press/v202/nikdan23a.html \
> [2] Mahdi Nikdan, Soroush Tabesh, Elvir Crn ˇcevi ´c, and Dan Alistarh. RoSA: Accurate parameter-
> efficient fine-tuning via robust adaptation. In Forty-first International Conference on Machine
> Learning, 2024. URL https://openreview.net/forum?id=FYvpxyS43U \
> [3] PyTorch is actively developing tools for sparse matrices, with the beta version available at https://pytorch.org/docs/stable/sparse.html \
> [4] Mohammad Hasanzadeh Mofrad, Rami Melhem, Yousuf Ahmad, and Mohammad Hammoud. Multi-threaded layer-wise training of sparse deep neural networks using compressed sparse column. In 2019 IEEE High Performance Extreme Computing Conference (HPEC), pp. 1–6. IEEE, 2019. \
> [5] Connor Holmes, Minjia Zhang, Yuxiong He, and Bo Wu. Nxmtransformer: semi-structured sparsifi-cation for natural language understanding via admm. Advances in neural information processing systems, 34:1818–1830, 2021. \
> [6] https://github.com/IST-DASLab/spops

---

### Author Response · Authors · 2024-11-26

**Rebuttal change summary:**

1. We added paragraphs in section1 (line 60-68, 87-89) and section 2 (line 123-137) to more clarify the challenge of sparse fine-tuning. We also added some words at the research questions (line 69-72).
2. We described more on the memory benefit of our approach in detail (line 209-211, re-wrote section 5.4, and added appendix D). We also added memory footprints to all experimental results of Llama.
3. We included the discussions and corresponding descriptions for LoRA’s variants (line 148-152, 214-215, 281-284).
4. We changed the setting of fine-tuning Llama using our approach to be similar settings compared with LoRA (line 315-316, Table 6, 9, 10).
5. We incorporated additional experiments (line 261-262, 309-310, 871-878, Table 2 and 4, and appendix D).
6. We rewrote, added texts, and added citations for some minor improvements (line 21-24, 76-77, 90-98, 269, 296-297, 322, 333-336, 459-470, 526-527, 800, and Equation 6).

Overall, in the revision, we included the memory footprint for Llama and additional experimental results. These results underscore the remarkable memory efficiency of our approach: even when the trainable parameters of our approach are 3 times that of LoRA, the memory usage of our method remains smaller than LoRA's.

We have also uploaded a **revision track** highlighting the revised text in red, as included in the **supplementary material**.


**Shared feedbacks for all reviewers:**

We thank all the reviewers for the effort and the valuable and constructive feedback. We “presented a parameter-efficient fine-tuning method inspired by techniques used in network pruning” (see reviewer 5q3j) and we “validate the proposed method with extensive experiments on both vision and language tasks” (see reviewer H135). Our approach “uses structured pruning and matrix decomposition to greatly reduce memory and parameter requirements while maintaining performance, suitable for scenarios with limited computational resources” (see reviewer nTJT). In the experimental results for Llama, we “provide a comprehensive comparison with existing state-of-the-art methods like LoRA, demonstrating improvements in memory efficiency methods and maintaining comparable accuracy, which is crucial for practical applications” (see reviewer H135). We also “add an additional, helpful section on the practical considerations of memory requirements when using the Adam optimizer” (see reviewer 3hdW).

The main concerns of the reviewers were related to argument and experimental comparisons. In our revision, we re-structured and re-wrote the text and added some experimental results to improve the argument. The modification is summarized in the rebuttal change summary. Then, we address each point individually below.

We will be happy to engage in further discussion.

---

### Comment · Area_Chair_Xtgc · 2024-11-28
**Reviewers please respond**

Dear Reviewers,

If you have not responded to author's rebuttal, please kindly do so as soon as possible. The deadline is Dec 2, but the authors can potentially further clarify questions if you respond earlier. Thanks!

Best,\
AC

---

### Meta-Review · Area_Chair_Xtgc · 2024-12-23

**Metareview:**

(a) Summary: a method for PEFT that only tunes weights on "important" weights, based on insights from neural pruning; experiments show improvement over existing sparse-fine-tuning methods.

(b) Strengths: experiments over various settings; higher efficiency in memory; practical method.

(c) Weaknesses: lack of comparisons with key LoRA variants; trailing performance with LoRA; more fine-grained downstream datasets result missing; missing details on memory efficiency

(d) reasons for decision
missing important experiments; efficiency not better than LoRA. Not enough support from reviewers.

**Additional Comments On Reviewer Discussion:**

Many concerns are addressed, or partially addressed, but concerns on missing datasets/experiments, and comparisons with LoRA/LoRA variants remain.

---

### Decision · Program_Chairs · 2025-01-22

Reject